# ALIGNED DATASETS IMPROVE DETECTION OF LATENT DIFFUSION-GENERATED IMAGES

**Anirudh Sundara Rajan*  Utkarsh Ojha*  Jedidiah Schloesser  Yong Jae Lee**

University of Wisconsin-Madison

{asundararaj2, uojha, jschloesser}@wisc.edu, yongjaelee@cs.wisc.edu

## ABSTRACT

As latent diffusion models (LDMs) democratize image generation capabilities, there is a growing need to detect fake images. A good detector should focus on the generative model's fingerprints while ignoring image properties such as semantic content, resolution, file format, etc. Fake image detectors are usually built in a data-driven way, where a model is trained to separate real from fake images. Existing works primarily investigate network architecture choices and training recipes. In this work, we argue that in addition to these algorithmic choices, we also require a well-aligned dataset of real/fake images to train a robust detector. For the family of LDMs, we propose a very simple way to achieve this: we reconstruct all the real images using the LDM's autoencoder, without any denoising operation. We then train a model to separate these real images from their reconstructions. The fakes created this way are extremely similar to the real ones in almost every aspect (e.g., size, aspect ratio, semantic content), which forces the model to look for the LDM decoder's artifacts. We empirically show that this way of creating aligned real/fake datasets, which also sidesteps the computationally expensive denoising process, helps in building a detector that focuses less on spurious correlations, something that a very popular existing method is susceptible to. Finally, to demonstrate the effectivenss of dataset alignment, we build a detector using images that are not natural objects, and present promising results. Overall, our work identifies the subtle but significant issues that arise when training a fake image detector and proposes a simple and inexpensive solution to address these problems. For implementation details, visit our project page: anisundar18.github.io/AlignedForensics/.

## 1 INTRODUCTION

There has been a transformation in the visual world of the internet with the rise of modern generative models. Diffusion models (Sohl-Dickstein et al., 2015; Song & Ermon, 2020; Ho et al., 2020; Dhariwal & Nichol, 2021) have been central to this shift, aided by internet-scale vision-language datasets like LAION (Schuhmann et al., 2022), allowing users to create images from text (Ramesh et al., 2022; Rombach et al., 2022; Saharia et al., 2022). Some welcome these tools for boosting creativity and productivity; others, less so. Recently, AI-generated images led Twitter users to falsely believe Donald Trump had been arrested[1]. As these algorithms grow more powerful, skepticism over online images rises. Consequently, the field of fake image detection has grown to try to address these issues.

Class-conditional pixel-space diffusion models (Saharia et al., 2022; Ramesh et al., 2022) are computationally demanding and difficult to customize. Latent Diffusion Models (LDMs) (Vahdat et al., 2021; Rombach et al., 2022) mitigate these challenges, making diffusion models more accessible. Recent open-source text-to-image models (Chen et al., 2023; Li et al., 2024; Podell et al., 2023) exemplify this shift, though the same efficiency also raises concerns about misuse. Existing works (Corvi et al., 2022; Cozzolino et al., 2024; Ojha et al., 2024) detect images from these generators

---

* Equal contribution

[1]https://x.com/EliotHiggins/status/1637927681734987777?s=20

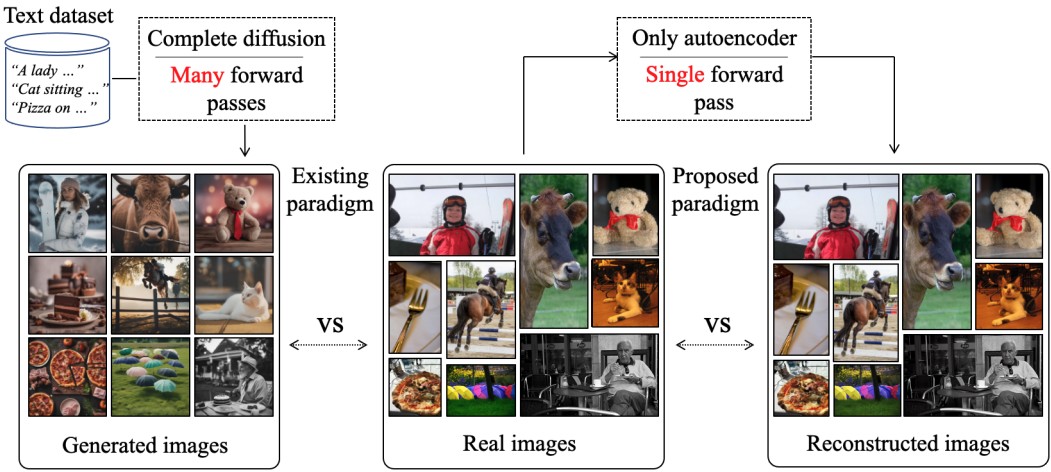

Figure 1: Different ways of generating images with the aid of latent diffusion models (LDMs). The most popular way (left) is to start from noise and a text prompt and go through the denoising process over many steps using a particular configuration (e.g., guidance scale, resolution, aspect ratio). Our proposed approach is to take a set of real images (middle) in their *original* form (e.g., aspect ratio) and reconstruct them using only the LDM's autoencoder (right) without the denoising process.

but often fail on post-processed ones, thereby limiting their practical use. Therefore, rather than focusing on generalization to unseen generators, we aim to improve the reliable detection of fake images from latent diffusion models (fully generated, not partially inpainted).

The most effective way to detect fake images involves two steps: collecting real images and generating fake ones, then training a classifier to distinguish them. While much research focuses on training strategies such as using augmentations (Wang et al., 2020) or minimizing trainable parameters (Ojha et al., 2024; Ricker et al., 2024)—the choice of real/fake dataset remains crucial. Ideally, the only difference between the real and fake images should be the artifacts introduced due to the generative model. We believe that there is room for improvement in this area.

For training a fake image detector, dataset design is of critical importance. During training, the detector could latch onto subtle differences between the real and fake images in the dataset. If these differences are not controlled, the detector could learn to focus on spurious patterns. We highlight some of these errors, and propose a principled way to mitigate these issues. Our key contribution for better real/fake alignment is very simple - we take a set of real images (Fig. 1 middle) and reconstruct them using the generative model (Fig. 1 right). These reconstructions are near-identical to their real counterparts visually- e.g., in resolution, semantic content and color tone, except that they have artifacts introduced through the generative model. Hence, they serve as our fake images. With this improved alignment, we observe that the detector focuses less on the false patterns. With latent diffusion models, we can use their VAE (Kingma & Welling, 2022; van den Oord et al., 2018) to reconstruct the real images. By doing so, we force the fake detector to focus on the fingerprints of the VAE decoder. Since all kinds of generated images always pass through the decoder, they must share the same fingerprints.

If we compare our method to the more common paradigm in which fake images are generated using the full denoising process (Ojha et al., 2024; Corvi et al., 2022; Cozzolino et al., 2024), there are many advantages. First, our way of getting the fake images for training is not as expensive. The standard way of generating images using diffusion models is an iterative, time consuming process. In our case, there is just one forward pass through the autoencoder. Second, in the standard setup, the real and fake images could have different properties beyond just the generative model's artifacts. For example, their resolutions could be different; and if they have to be resized to a fixed size, real and fake images could have different amount of resizing artifacts, something that the classifier can latch on to during training. We show that one of the most effective detectors (Corvi et al., 2022) indeed suffers from this problem, and how our better aligned training data results in a much more robust detector. Third, since generating fake images aligned to a set of real images is so simple using our method (we don't have to worry about setting the ideal guidance scale or prompt engineering), we can use any set of real images to train the detector. To demonstrate this point,

we collect algorithmically generated images using OpenGL, which look nothing like natural images (Baradad et al., 2023). Treating those as our real images, we obtain their reconstructions and train a detector to distinguish the two. The detector works well not just in detecting other algorithmically generated real/fake images, but also in detecting natural looking real/fake images. This unreasonable effectiveness points to a principle about building robust detectors: it may be more important how real and fake images differ from each other, and less important how real images in themselves are.

We conduct extensive experiments to assess our method. We train on images generated by the original LDM model (Rombach et al., 2022), and test on images generated by later versions of Stable Diffusion as well as newer latent models such as Playground (Li et al., 2024), Kandinsky (Razzhigaev et al., 2023), PixelArt-$\alpha$ (Chen et al., 2023) and Latent Consistency models (Luo et al., 2023). We also test it on images generated by Midjourney (mid), a closed-source commercial model. Our method is able to match, and often outperform, the best existing detectors in detecting fake images. Using our approach, one can train a detector that is less-likely to focus on spurious patterns. Finally, since we generate fake training images using the LDM's autoencoder, without any denoising operation, our approach is 10x less-expensive than state-of-the-art methods. We also identify and explain the limitations of our approach. Overall, we hope that our work highlights the subtle errors that can occur when training fake image detectors and encourages further research on training detectors without these errors.

## 2 RELATED WORK

Several algorithms to identify fake images have been proposed. Wang et al. (2020) fine-tunes a ResNet-50 (He et al., 2015) model to classify images as either real or fake. It trains on fake images generated by ProGAN (Karras et al., 2018). Training with aggressive data augmentation enables the detector to successfully identify images generated by various CNN-based models. To better preserve low-level details of the image, Gragnaniello et al. (2021) removes the downsampling operations present in the initial layers of the neural network. Furthermore, Chai et al. (2020) trains a detector on image patches to force the model to learn local fingerprints. Combining the above practices, Corvi et al. (2022) trains a fake image detector for LDM generated images. Unlike previous methods that fine-tune the entire neural network, Ojha et al. (2024) demonstrate that linear probing of a pre-trained, frozen CLIP image encoder (Radford et al., 2021) can effectively detect fake images generated by a wide range of models. Work by Cozzolino et al. (2024) extends this approach to detect images generated by additional models. Despite these algorithmic advances, the real and fake images used for training these models are not well-aligned. Our work demonstrates the various benefits that come from creating aligned fake images that are reconstructions of real images.

The use of reconstructions for fake image detection has been explored in prior work. Chai et al. (2020) create a dataset of GAN-generated fakes by reconstructing real images (Bau et al., 2019), but find that detectors trained this way underperform compared to those trained on randomly generated images—likely due to reconstruction inaccuracies. The most similar work to ours is DRCT (Chen et al.), which reconstructs real and fake images using DDIM inversion. However, their dataset remains misaligned since the fake images lack corresponding real counterparts. Moreover, DRCT depends on the costly and time-consuming DDIM inversion, while we show that simple VAE reconstructions suffice for training an effective LDM-image detector. In contrast to these works, we use VAE reconstructions that efficiently preserve real image quality while outperforming detectors trained on iteratively denoised images.

Reconstruction-based detection has also been studied for diffusion models. Wang et al. (2023) use DDIM inversion (Song et al., 2022) to reconstruct images, training a classifier on their difference (DIRE). Ricker et al. (2024) extend this to LDMs, using VAE-based reconstructions and LPIPS distance (Zhang et al., 2018) for detection. However, these methods require access to the generative model at inference and fail under common post-processing. In contrast, our approach uses reconstructions only during training, enabling faster inference and greater robustness to corruptions.

## 3 PRELIMINARIES

Our goal is to build a fake image detector to detect whether an image is real or fake, i.e., created with the aid of a generative model. We first explain our problem statement and then provide an overview of the existing paradigm.

### 3.1 PROBLEM SETUP

Recently, various image generation models have emerged. Among them, latent diffusion models (LDMs) (Vahdat et al., 2021; Rombach et al., 2022) and the subsequent stable diffusion (SD) series have become noticeably ubiquitous. This is because they strike a good balance: users get control (e.g., through text prompts) to generate quality synthetic images and the overall process is computationally more efficient than other text-to-image diffusion models (Saharia et al., 2022; Ramesh et al., 2022). Hence, our specific goal in this work is to detect images generated by LDMs.

### 3.2 LATENT DIFFUSION MODELS

A latent diffusion model first compresses images into a lower dimensional latent space using an autoencoder. The diffusion model then operates in this latent space. A text-conditioned latent diffusion model consists of an encoder ($\phi_{enc}$), decoder ($\phi_{dec}$) and a UNet ($\epsilon_\theta$). Given a high dimensional image space $\mathcal{X}$ and a lower-dimensional latent space $\mathcal{Z}$, we train an encoder network to compress an image ($x \in \mathcal{X}$) into a lower dimensional latent ($z \in \mathcal{Z}$). The decoder is trained to reconstruct the image from the latent.

In order to generate an image based on a conditioning signal $c$ (prompt, bounding box, reference image, etc.), we first sample $z_T$ from a fixed prior (gaussian noise), where $T$ corresponds to the number of timesteps. We then denoise the image for $T$ steps using the UNet to get a latent $z_0$. We pass $z_0$ through the decoder $\phi_{dec}$ to generate the image.

### 3.3 EXISTING FAKE DETECTION PARADIGM

Since we don't know what precisely makes a fake image *fake* and a real image *real*, a common and effective approach has been to learn that difference (Wang et al., 2020; Corvi et al., 2022; Cozzolino et al., 2024). The first step is to create a dataset consisting of two categories (i) $\mathcal{R}$ consisting of real images and (ii) $\mathcal{F}$ consisting of fake images sampled using a generative model $\mathcal{G}$. The next step involves training a deep neural network $\psi$ on the collected dataset $\mathcal{D} = \{\mathcal{R} \cup \mathcal{F}\}$ for the task of binary classification, so that $\forall x \in \mathcal{R}, \psi(x) \approx 0$ and $\forall x \in \mathcal{F}, \psi(x) \approx 1$. The hope is that when $\psi$ is learning to separate the distribution of $\mathcal{F}$ from $\mathcal{R}$, it does so *only* by discovering $\mathcal{G}$'s artifacts in $\mathcal{F}$ (something not present in $\mathcal{R}$), and not by using some other spurious features. We explain what these spurious features can be with the help of an example prior work.

**Imperfect alignment between $\mathcal{R}$ and $\mathcal{F}$:** The work of Corvi et al. (2022) trains a detector in the same way discussed above. For the real data $\mathcal{R}$, it uses MS COCO (Lin et al., 2015) and LSUN (Yu et al., 2016). For fake images $\mathcal{F}$, it uses the text prompts from MS COCO to generate images from an LDM model at a fixed 256 x 256 resolution. The real images tend to be at a higher resolution than fake images (see Appendix A.1.1 for details). Fig. 1 (left) shows the discrepancy in the sizes of real and fake images. To make the detector robust to image resizing, the random resized crop [2] data augmentation is used during training. By first cropping the image and then resizing the cropped image to a fixed final resolution, the random resized crop introduces both upsampling and downsampling artifacts to the training data. However, due to the discrepancy in resolution between real and fake images, the random resized crop introduces different signals to each distribution. Consequently, there is potential for $\psi$ to use this signal in some capacity to separate $\mathcal{R}$ from $\mathcal{F}$. Similar issues have been illustrated in prior works from Grommelt et al. (2024); Yan et al. (2024). Later on in Sec. 5, we show that the resulting detector indeed learns such spurious features and changes its prediction if the same image is saved at a different resolutions. We want to avoid such situations and instead look for a principled way to learn a robust fake image detector.

### 4 OUR APPROACH

If we do not want $\psi$ to learn any spurious features, then we need to make sure they are not available for $\psi$ to learn from the training data itself. So, our key idea is to align $\mathcal{R}$ and $\mathcal{F}$ as much as possible so that their only difference is due to $\mathcal{G}$'s artifacts. Note that prior methods (Ojha et al., 2024; Corvi et al., 2022; Cozzolino et al., 2024) do try to align $\mathcal{R}$ and $\mathcal{F}$ in some capacity. For example, while

---

[2]https://pytorch.org/vision/stable/generated/torchvision.transforms.RandomResizedCrop.html

generating $\mathcal{F}$, instead of using arbitrary prompts, they use $\mathcal{R}$'s image descriptions to generate images using LDM. However, such images are only similar to the real images in terms of semantics. We instead propose an approach that can bring the distribution of $\mathcal{F}$ to be much closer to that of $\mathcal{R}$, which can aid in subduing many of the other differences in image properties (e.g., aspect ratio).

Our solution is simple. To generate $\mathcal{F}$, we reconstruct images in $\mathcal{R}$ using $\mathcal{G}$'s parameters. Typically, reconstructing an image $x$ involves a multi-step inversion (e.g., DDIM inversion (Song et al., 2022)) from the image space to the noise space to compute latent $z_x$, such that $\mathcal{G}(z_x) \approx x$. However, for the particular $\mathcal{G}$ that is our target in this work, i.e., LDM/SD, there is a much simpler solution. Specifically, given a real image $x$, we pass it only through the pre-trained autoencoder, without using the U-Net to do any forward/reverse process. Using the notations defined in Sec. 3.2, we can mathematically formulate the process in the following way:

$$\mathcal{F} = \{ \, \phi_{dec}(\phi_{enc}(x)) \quad | \quad \forall x \in \mathcal{R} \}$$

The resulting reconstructed images can still be validly considered fake, at least for our task, since they necessarily contain artifacts introduced by $\phi_{dec}$. We create this distribution of $\mathcal{F}$ from $\mathcal{R}$ before any training begins. Once we have this data, we then train a deep neural network $\psi$ using the same training recipe as proposed in Corvi et al. (2022). Specifically, we use ResNet-50 (He et al., 2015) pretrained on ImageNet and finetune the whole backbone for real-vs-fake image classification. In each iteration, we sample a batch of real $\mathcal{R}_i$ and fake images $\mathcal{F}_i$, and train $\psi$ using binary cross entropy loss (real: 0 and fake: 1).

Within this framework, we consider two variants depending on the composition of real and fake images in a batch ($i$). In one case, the batch could be a random assortment of real and fake images where images in $\mathcal{R}_i$ and $\mathcal{F}_i$ do not need to have any correspondence between them. In the other case, each image in $\mathcal{F}_i$ has a corresponding real image in $\mathcal{R}_i$ as part of the same batch, with identical data augmentations (e.g., crop). With this latter variant, the alignment between real and fake images is ensured not just at the dataset level, but also at the batch level in each iteration, and we explore if this further helps the model to focus on the desired features for fake detection. We call the two variants *Ours* and *Ours-Sync* respectively. For both, since we train $\psi$ to focus on the artifacts of $\phi_{dec}$, the detector should detect even those images which have been produced through the full denoising process using the U-Net $\epsilon_\theta$. This is because even those images will have their denoised latents go through the same decoder $\phi_{dec}$, and hence should have similar artifacts.

## 5 EXPERIMENTS

In this section, we demonstrate that using reconstructions provides a very inexpensive way to create a well-aligned real-vs-fake training dataset which in turn reduces the likelihood of a detector learning spurious patterns. Our experiments show that a detector trained using our approach can effectively detect fake images generated by various other text-to-image latent diffusion models. Finally, we show that as long as the real-vs-fake dataset is well-aligned, the content of the images is relatively less significant for training a fake image detector.

### 5.1 BASELINES AND TRAINING DETAILS

We consider the following baselines for fake image detection: (i) CLIP-based detection (Ojha et al., 2024), using linear probing on real/fake images with variants trained on GAN (Ojha-ProGAN) and LDM (Ojha-LDM) data. (ii) Cozzolino-LDM (Cozzolino et al., 2024), which improves dataset quality and leverages an improved CLIP backbone. (iii) AEROBLADE (Ricker et al., 2024), a training-free method based on LDM autoencoder reconstructions. (iv) Corvi (Corvi et al., 2022), a ResNet-50 trained on real (MS COCO + LSUN) and LDM-generated images, combining augmentation-driven (Wang et al., 2020) and patch-based training (Chai et al., 2020).

We dedicate the first part of our experiments to comparing our method exclusively to (Corvi et al., 2022). There are two reasons. First, Corvi is one of the most effective methods for detecting fake images from various text-to-image latent diffusion models (we compare all methods later in Table 1). Yet, the training pipeline used by Corvi gives rise to the classifier learning some spurious features, and hence is a good test bed for our method, to see whether we can avoid making our detector learn those features, while still preserving or improving fake detection in a general sense. Second, since Corvi represents the existing paradigm of collecting fake images for training through

a computationally intensive iterative denoising process, we compare our approach to assess the increased efficiency of our dataset creation method.

**Training details:** Similar to Corvi et al. (2022), we use a combination of MS COCO (Lin et al., 2015) and LSUN (Yu et al., 2016) as our real dataset, totaling 179257 images. We reconstruct them using the autoencoder of the LDM model proposed by Rombach et al. (2022) to get the same number of fake images. Starting from ImageNet pretrained ResNet-50 as $\psi$, we finetune it on the real-vs-fake dataset. We optimize using Adam (Kingma & Ba, 2015) with an initial learning rate set to 0.0001. The rest of the training details can be found in Appendix A.1.1.

## 5.2 SUBDUING SPURIOUS FEATURES

In our work, we emphasize that a well-aligned real-vs-fake dataset reduces susceptibility to spurious features. Here, we test this by analyzing the detector's sensitivity to image resizing. We also study the sensitivity of our detector to other post-processing operations in Appendix A.2.5.

**Issue caused by lack of well-aligned data:** Corvi et al. (2022) use LDM-generated images as fake and LSUN + MS COCO as real. All fake images are 256×256, while LSUN images match this resolution. However, most COCO images are significantly larger, making real images, on average, higher resolution than fake ones. A key component in the data augmentation pipeline is the 'random resized crop' function, which works in the following way: a random crop is taken as a percentage of the whole image. This percentage is chosen uniformly from the range [8,100]. The resulting crop is then resized to a fixed 256 x 256 resolution. Hence, resize cropped fake images have *only* up-scaling artifacts. On the other hand, many real images will have both up-scaling and down-scaling artifacts. In contrast, our method uses the same real images (COCO + LSUN), but our fake images are reconstructed from real ones, preserving their exact resolution. While we fine-tune the network similarly to Corvi, we show that these issues also affect other training-based methods (e.g., Ojha-LDM). See Appendix A.2.2 for a detailed analysis.

**Experiment details:** We create a test set of real and fake images of increasing/decreasing resolutions. For the real set, we randomly select 500 images from the Redcaps dataset (Desai et al., 2021). For fake images, we generate 500 images using SD 1.5 (prompts pertain to object categories from CIFAR (Krizhevsky et al., 2010)). The original resolution of both real and fake images is 512 x 512. We resize these images to different resolutions in the range of [128, 1024] corresponding to scaling factors of 0.25 and 2.00 respectively. We then test both detectors, Corvi and ours, to see how they perform on this dataset. Both methods produce the probability of an input image being fake.

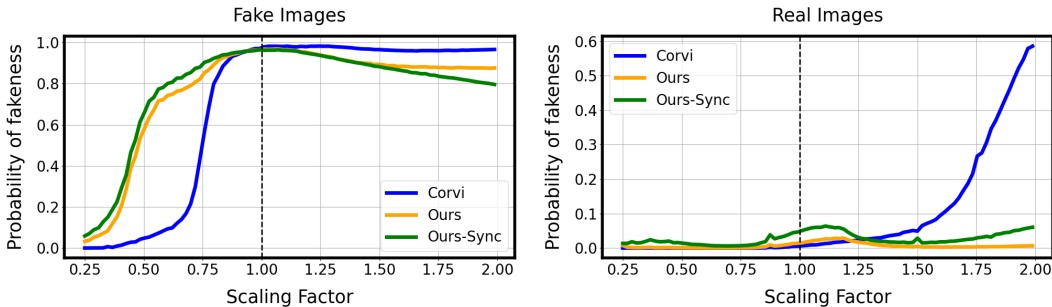

Figure 2: **Sensitivity of fake detectors to image resizing** for a set fake images (left) and a set of real images (right). Corvi associates downsampling with real images and upsampling with fake images. Our detectors do not learn that false pattern, showing better robustness.

**Results and analysis:** In Fig. 2 left and right, we see the behavior of the two detectors on fake and real images respectively, where x-axis denotes the scaling factor of test images, and y-axis represents the fakeness score of the model. We first focus on the performance on fake images (left). Soon after we start downsizing from the base resolution (marked with the dotted line), the probability of the image being fake drops drastically. This makes sense since during training, the Corvi detector has only seen downsizing artifacts on real images. Our method's performance declines much more gradually, struggling only under extreme downsampling. We attribute this drop to the lack of scale invariance in CNNs. Similarly, when upsampling real images (Fig. 2 right), the performance of

|  | Real | SD | MJ | Kandinsky | Playground | PixelArt-$\alpha$ | LCM |
|---|---|---|---|---|---|---|---|
| AEROBLADE (Ricker et al., 2024) | 96.58 | 74.50 | **99.50** | 99.26 | 14.87 | 75.93 | 99.96 |
| Ojha-ProGAN (Ojha et al., 2024) | 95.13 | 17.84 | 12.96 | 23.56 | 21.03 | 19.73 | 23.93 |
| Ojha-LDM (Ojha et al., 2024) | 54.16 | 69.56 | 69.40 | 90.70 | 92.16 | 90.73 | 73.66 |
| Cozzolino-LDM (Cozzolino et al., 2024) | 85.36 | 47.36 | 50.93 | 51.06 | 59.73 | 59.52 | 34.70 |
| Corvi (Corvi et al., 2022) | **99.96** | **99.73** | 96.90 | **99.92** | 82.13 | **100** | 99.60 |
| *Ours* | 99.93 | 99.31 | 98.50 | **99.92** | 94.85 | **100** | **100** |
| *Ours-Sync* | 99.76 | 99.57 | 99.37 | 99.57 | **99.48** | **100** | **100** |

Table 1: **Generalization results**. Accuracy of different methods for detecting real and fake images. The finetuning based detectors (Corvi, *Ours*, and *Ours-Sync*) generally outperform the training-free and linear probing based methods. Our approach is robust to drastic changes in the UNet architecture (Playground) showing a huge improvement (+12.72/+17.35 for Ours/Ours-Sync) from the Corvi detector.

the baseline detector worsens since during training, up-sampling artifacts are seen more with fake images. Our detectors remain much more consistent on real images throughout the resizing range. Overall, these results highlight the effectiveness of the simple way in which we can make the detector much more robust through a better dataset alignment.

### 5.3 EVALUATION ON DIFFERENT TYPES OF REAL/FAKE IMAGES

Now that we have seen the benefits of our method in being able to avoid certain spurious correlations while being computationally very efficient, we now study how it fares in being able to detect different types of fake images produced by different types of latent diffusion models.

**Test datasets**: We compare all the detectors introduced in Sec. 5.1 to our method on the following datasets. (i) The *Real* set contains real images from multiple sources; 1000 images from Red-Caps (Desai et al., 2021), 800 images from LAION-Aesthetics (Schuhmann et al., 2022), 1000 images from whichfaceisreal (whi) and 200 images from WikiArt (wik). For fake images, we collect images from the following sources. (ii) Different variants of Stable Diffusion (*SD*), which includes 1000 fake images from InstructPix2Pix (Brooks et al., 2023), 1000 images from Nights (Fu et al., 2023) dataset and 1000 DDIM inversion of real face images; (iii) 3000 images from Midjourney (*MJ*) (mid), whose model architecture is not public; (iv) 3000 images from *Kandinsky* (Razzhigaev et al., 2023) which has a VAE of a different architecture in comparison to the LDM model we train on; (v) 3000 images from *Playground* (Li et al., 2024) and (vi) 3000 from *PixelArt-$\alpha$* (Chen et al., 2023), which have similar VAEs as ours, but their U-Net is different; and (vii) 3000 images from *Latent consistency* model (LCM) (Luo et al., 2023) which was distilled from a finetuned version of SD 1.5 using the objective proposed by Song et al. (2023).

We create this overall test set to ensure that there is enough diversity of natural and artistic looking images. The selected models offer a wide range of architectural choices utilized by latent diffusion models. Furthermore, for these same real/fake images, we separately construct their post-processed versions where we randomly add JPEG compression, blurring, color jitter, and resize each image. This enables us to test the robustness of detectors to common post-processing operations. Following (Ojha et al., 2024), we use accuracy as the evaluation metric with 0.5 as the threshold (threshold details for AEROBLADE can be found at Appendix A.1.3). We also report threshold-less metrics such as average precision in Appendix A.2.3.

**Results and analysis**: Tables 1 and 2 show the performances of various baselines on our test set, before and after applying common post-processing operations, respectively. Overall, the methods that involve fine-tuning (Corvi and Ours) outperform the other methods that are either linear probing based (Ojha-ProGAN, Ojha-LDM, Cozzolino-LDM) or without training (AEROBLADE). Ours further outperforms Corvi, especially on post-processed images (Table 2). Both Corvi and our method use the common image corruptions as data augmentations during training, however our method shows huge improvements; e.g., our approach obtains a +36.98/+52.09 improvement for Ours/Ours-Sync over Corvi on Playground generated images. Playground is built from SDXL (Podell et al., 2023), which uses a fine-tuned version of the LDM autoencoder, but uses a UNet with thrice as many parameters compared to LDM. We hypothesize that since our method only focuses on the VAE, it is robust to drastic changes in UNet architecture as opposed to Corvi which uses the UNet for training.

| | Real | SD | MJ | Kandinsky | Playground | PixelArt-$\alpha$ | LCM |
|---|---|---|---|---|---|---|---|
| AEROBLADE (Ricker et al., 2024) | 96.85 | 28.10 | 61.80 | 27.60 | 3.17 | 10.22 | 4.60 |
| Ojha-ProGAN (Ojha et al., 2024) | 93.26 | 13.10 | 8.26 | 14.53 | 7.63 | 9.60 | 13.16 |
| Ojha-LDM (Ojha et al., 2024) | 48.46 | 57.20 | 51.63 | 64.63 | 66.16 | 64.37 | 64.33 |
| Cozzolino-LDM (Cozzolino et al., 2024) | 76.63 | 52.80 | 59.43 | 59.63 | 73.36 | 66.22 | 60.93 |
| Corvi (Corvi et al., 2022) | 98.40 | 77.78 | 50.45 | 59.23 | 24.27 | 64.41 | 58.31 |
| *Ours* | **99.85** | 86.50 | 70.68 | 64.88 | 61.25 | 84.36 | 90.12 |
| *Ours-Sync* | 99.50 | **88.06** | **72.36** | **68.60** | **76.36** | **86.10** | **91.08** |

Table 2: **Sensitivity to common post-processing operations**. Analogous to Table 1, but the test images have undergone random compression, resizing, blur and color jitter. Our detectors show improved robustness over the baselines when detecting fake images from different latent diffusion models. Ensuring batch-level alignment (*Ours-Sync*) offers increased robustness.

Like Corvi, the other baselines also struggle with architectural differences used to process training data vs. testing data. For example, AEROBLADE uses the VAEs of SD 1.1, SD 2 and Kandinsky 2.1 and uses the smallest reconstruction error of the three to perform real/fake detection. It is unable to detect images from SD, Playground, and PixelArt-$\alpha$ as they use a different VAE. Furthermore, AEROBLADE struggles to detect images that have been through common post-processing operations. Interestingly, the Kandinsky VAE uses a different architecture from the LDM VAE that we trained on, as it replaces the convolutional decoder of LDM with a MoVQ decoder (Zheng et al., 2022) for improved generation quality. We observe that our detector is less sensitive to this change than other methods, likely because the other methods additionally utilize the UNet or are more susceptible to spurious correlations due to not enforcing aligned data during training, leading to larger distributional differences in the generated training vs. testing images. Finally, *Ours-Sync* surpasses *Ours*, suggesting that batch-level alignment during training improves generalization.

Our results confirm the advantage of using well-aligned real/fake training images from the LDM's VAE. This helps the detector focus on genuine signals, reduces spurious correlations, and improves robustness to UNet changes. While we use a ResNet-50 backbone with ImageNet initialization, Appendix A.2.4 shows these findings hold across various architectures and initializations.

## 5.4 EFFECT OF DATASET SIZE

We study the effect of training dataset size on our methods and Corvi. Instead of training on all 179,257 images, we try training each method using smaller dataset sizes. We test by sampling of 1000, 10,000, 50,000 and 100,000 real and fake images each from our training distribution and report the results here. We evaluate the detectors based on their performance on the whole test dataset from Section 5.3. Our real distribution has 6000 images, consisting of both the original and post-processed real images. We have 30,000 images in our fake distribution coming from the 6 models that we test on. Furthermore, in order to disentangle the effects of resizing, we do not use resizing as part of post-processing here. Due to the imbalanced nature of the dataset, we report the true positive rate (TPR) at a fixed false positive rate of 5%.

| **Dataset Composition** | Corvi | *Ours* | *Ours-Sync* |
|---|---|---|---|
| 1k / 1k | 46.51 | **83.37** | 80.43 |
| 10k / 10k | 87.64 | 98.13 | **98.58** |
| 50k / 50k | 93.27 | 99.56 | **99.81** |
| 100k / 100k | 95.84 | 99.69 | **99.75** |

Table 3: **Effects of varying the dataset size**. We report the TPR@5FPR for detectors trained on lesser data. Using an aligned dataset helps the model learn a good hypothesis with less data. *Ours/Ours-Sync* shows improvements of +36.86/33.92, +10.49/10.94, +6.29/6.54 and +3.85/3.91 over Corvi when trained with 1k, 10k, 50k and 100k images (each) respectively.

We report our results in Table 3. When training on a small dataset size of 1k real and fake images each, the Corvi detector can detect only 46.51% of the fake images while allowing having a 5%

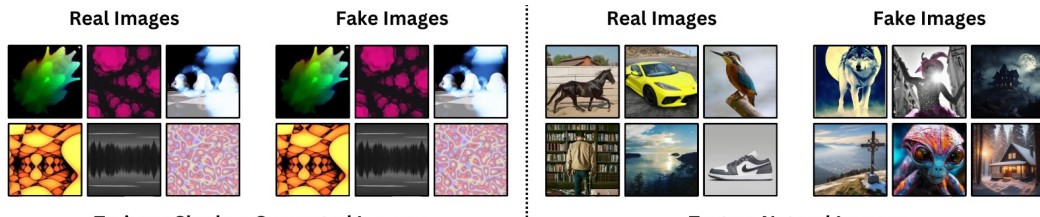

Figure 3: We use OpenGL shader generated images (Baradad et al., 2023), as our real images and reconstruct them to obtain our fake images. We then train a detector using this dataset.

| | Original | | | | With Post-Processing | | | |
|---|---|---|---|---|---|---|---|---|
| | Corvi | *Ours-Sync* | *Ours (shaders)* | *Ours-Sync (shaders)* | Corvi | *Ours-Sync* | *Ours (shaders)* | *Ours-Sync (shaders)* |
| Real | **99.91** | 99.73 | 84.98 | 83.24 | 97.85 | **98.86** | 74.62 | 76.62 |
| SD | 99.68 | **99.80** | 99.75 | 99.44 | 78.05 | **84.73** | 77.82 | 72.30 |
| MJ | 97.71 | **99.30** | 98.56 | 95.79 | 51.03 | **72.38** | 53.45 | 46.43 |
| Kandinsky | **99.88** | 99.80 | 98.23 | 98.10 | 58.68 | 69.02 | **69.29** | 67.87 |
| Playground | 86.25 | 98.80 | 99.82 | **99.90** | 24.11 | **63.83** | 52.65 | 54.66 |
| PixelArt-$\alpha$ | **100** | **100** | **100** | **100** | 63.34 | **80.99** | 69.65 | 68.81 |
| LCM | 99.65 | **99.93** | 99.78 | 99.84 | 50.63 | **82.83** | 63.14 | 60.81 |

Table 4: **Generalization results**. Comparing the model trained using the shaders dataset (Baradad et al., 2023) to models trained on natural image datasets (Corvi, *Ours-Sync*). All models use a threshold calibrated using a validation set. The detectors trained on shader images, show good performance on natural images. This shows that a well-aligned dataset is more important than the exact content of the images themselves.

false positive rate. *Ours* and *Ours-Sync* on the other hand, can detect 83.37% and 80.43% of the fake images respectively at the same threshold. A similar pattern can be seen with dataset sizes of 10k, 50k and 100k images each, this shows that Corvi needs a large dataset size in order to learn the correct hypothesis. As oppposed to ours, which can learn it in a data-efficient manner. Creation of the training dataset is also more efficient, which we show in Appendix A.2.1.

## 5.5 CAN WE TRAIN ON IMAGES THAT ARE NOT NATURAL?

Our experiments show that a properly aligned dataset can reduce spurious patterns, but both the training set and test settings still contain similar real-world concepts. But if the goal of alignment is to force the model to not look at *anything* else but the LDM decoder's artifacts, can the detector learn those artifacts without being trained on any *naturally occurring* real images at all? To study this, we next train our detector using a dataset which does not capture the semantic concepts that we test the model on, and see if it succeeds in detecting the same real/fake images described in Tables 1 and 2.

A similar motivation was discussed in Baradad et al. (2023), where the authors wanted to learn image representations without using real-world images. They proposed to generate images with 21,000 OpenGL fragment shaders, which are short programs that compute the color and transparency of every pixel in an image. Sample images are shown in Fig. 3 (left). Although the images are generated algorithmically, they do not involve a neural network. For our use case, we use these images as the "real" set. Specifically, our $\mathcal{R}$ consists of 100k such images of 384 x 384 resolution. We pass these images through the SD 1.5's VAE in order to get our reconstructions which serve as our fake distribution $\mathcal{F}$. We train the two versions of our detectors (normal and sync) in the same way as described in Sec. 4 (e.g., using same ResNet-50 as $\psi$), and evaluate them on the same test set from section 5.3. We train detectors using 5 random seeds and report the average values. Since, the model does not observe any natural images during training, it is not likely for those images to conform to a 0.5 threshold, therefore, we calibrate the threshold using the same validation set. For further details, we refer the reader to Appendix A.1.2.

**Results and Discussion**: We report accuracy of our shaders-trained detectors in Table 4. We also compare to detectors trained on natural real/fake images (Corvi and our method). For uniformity, we

train them using 100k real/fake images. First, we notice that *Ours (shaders)/Ours-Sync (shaders)* detect 84.98/83.24% of real images, compared to Corvi/*Ours-Sync* which detects 99.91/99.73% of real images. However, what is surprising is how effective our shaders-detectors are at being able to detect all types of fake images without perturbation, having almost the same accuracy as our detectors trained with natural images. If we especially compare them to Corvi on post-processed fake images (Table 4) (right), we see that they have a much better accuracy in almost every case by a decent margin; e.g., *Ours (shaders)* shows improvements of +10.61, +28.54, +6.31 and +12.51 on Kandinsky, Playground, PixelArt and LCM respectively. However, while *Ours-shaders* detectors do match the performance of our natural image trained detector (*Ours-Sync*) on clean fake images, they cannot do so on post-processed fake images. Even so, given that the detectors proposed in this section are trained without ever training on natural real image or a properly generated (iteratively denoised) fake image, being able to outperform the best existing detector (Corvi) which has both of those things highlights the crucial role that dataset alignment can play in deploying robust detectors.

# 6 DISCUSSION AND LIMITATIONS

Aligning data using the LDM's autoencoder assumes that most properties of real images can be transferred to the reconstructed image. However, there might be some low-level properties that do not get transferred as effectively. As a specific example, if real images are originally saved in .webp format, we find that the reconstructions might not inherit those compression artifacts, and hence the resulting detector is not completely robust to .webp compression. This is depicted in Fig 4, which plots the effect of different levels of .webp compression (x-axis) on the model's output score, i.e., probability of the image being fake (y-axis). We start with a clean set of 500 images generated by SD 1.5 (compression quality = 100). Both *Ours-Sync* as well as Corvi, which are trained on real images containing .webp artifacts, correctly assign a high score to the images. As we increase the compression level, the models' scores drop drastically. This implies that the detector has learned to associate .webp artifacts to real images. This is problematic as real images may contain unknown arbitrary properties, like .webp artifacts, which could make the detector sensitive to those features.

However, algorithmically generated images, like those discussed in Sec. 5.5, offer the advantage of complete control over their creation, allowing us to eliminate unnecessary artifacts. In fact, the detector trained on shaders generated images, *Ours (Shaders)*, is much more robust to .webp compression.

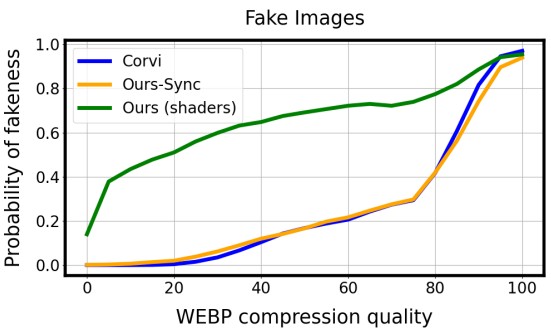

Figure 4: Sensitivity to webp compression

Finally, while we have shown that our method can be robust to small architectural changes in the U-Net (e.g., *Playground*) and VAE (*Kandinsky*), it struggles when there are major architectural differences in the VAE. As an example, we tested our detector on 1950 images generated by FLUX.1-dev (Labs), which utilizes a latent space of 16 channels (most SD variants have 4). The accuracy of *Corvi*, *Ours*, *Ours-Sync* in detecting them as fake are 3.18%, 9.59%, 25.87%. Perhaps unsurprisingly, this means that models with vastly different architectures tend to produce very different kinds of artifacts.

# 7 CONCLUSION

In this work, we demonstrated the need to train fake image detectors using a completely aligned dataset. We introduced a principled way to achieve this for Latent Diffusion Models. Through careful experimentation, we supported our claims. By training a fake image detector using OpenGL-generated texture images, we demonstrated the importance of focusing on the differences between the real and fake images, as opposed to the images themselves. An interesting future direction could be to study the application of this idea in the context of pixel-space diffusion models where the VAE is not available. We hope that our work highlights the importance of dataset alignment and paves the way for robust fake image detectors that can help society combat misinformation.

## 8 REPRODUCIBILITY STATEMENT

We provide precise details on our experimental setup to ensure reproducibility. For our experiments on training with natural-looking real images, we provide training details in Section 5.1 and Appendix A.1.1. Details of our test dataset can be found in Section 5.3. Details regarding our experiments on shaders can be found in Section 5.5 and Appendix A.1.2. We also intend to release our pre-trained checkpoints, datasets, and code to ensure reproducibility, with all resources made publicly available on GitHub.

## 9 ETHICS STATEMENT

During our study, we made sure to only use publicly available data, respecting people's privacy online. We didn't collect or analyze any personal or sensitive information. We believe our research can help tackle the spread of misinformation on the internet, contributing to a safer and more trustworthy digital space.

## 10 ACKNOWLEDGMENTS

This work was supported in part by NSF IIS2404180, American Family Insurance, Institute of Information & communications Technology Planning & Evaluation(IITP) grants funded by the Korea government(MSIT) (No. 2022-0-00871, Development of AI Autonomy and Knowledge Enhancement for AI Agent Collaboration) and (No. RS2022-00187238, Development of Large Korean Language Model Technology for Efficient Pre-training), and Microsoft Accelerate Foundation Models Research Program.

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

## A  APPENDIX

### A.1  IMPLEMENTATION DETAILS

#### A.1.1  ADDITIONAL TRAINING DETAILS

We follow the training recipe used by Corvi et al. (2022). We train on 96 x 96 crops of the whole image using a batch size of 128. The data augmentations include random JPG compression and blur from the pipeline proposed by Wang et al. (2020). Following Gragnaniello et al. (2021), grayscale, cutout and random noise are also used as augmentations. Finally, in order to make the network invariant towards resizing, the random resized crop was added. For our method as well as Corvi, we train the model using two different random seeds and report the average reading.

We use the validation set provided by Corvi et al. (2022) for our training. Just like our training set, the real images come from COCO/LSUN and the fake images are generated at 256 x 256 using LDM. During training, if the validation accuracy does not improve by 0.1% in 10 epochs the learning rate is dropped by 10x. The training is terminated at learning rate $10^{-6}$.

**Dataset details:** We use the dataset provided by Corvi et al. (2022). Half of the real images come from LSUN (Yu et al., 2016), the remaining comes from COCO (Lin et al., 2015). There are a total of 90,000 images from COCO out of which most of them are high resolution images. The most frequently occurring resolutions are 640 x 480 and 640 x 427 which occur 19,581 and 11,292 times respectively. For further details, we refer the reader to the download link[3] provided by Corvi et al. (2022).

#### A.1.2  SHADERS EXPERIMENT DETAILS

**Dataset details:**    All our images are at a resolution of 384 x 384. By inspecting the code used by Baradad et al. (2023), we figure out the exact post-processing done on our images. They were saved in the JPG format. Therefore, we also save our fake images in the JPG format ($quality \sim [70, 100]$) to make sure that the detector does not focus on compression artifacts.

---

[3]https://github.com/grip-unina/DMimageDetection/tree/main/training_code

**Training and Evaluation details:** We train our detectors using the configurations from before (refer Appendix A.1.1). In addition, we use the same validation set to compute the threshold for classification. It is important to notice that our validation set consists of natural looking images. We take 5000 real and fake images each from the validation set, and apply compression, resizing, blur and color jitter operations to these images. We evaluate all the models listed in Table 4 on the validation set, selecting the threshold for each model that achieves the best accuracy. We use this threshold when evaluating the model on the test set.

### A.1.3 EVALUATION OF AEROBLADE

AEROBLADE (Ricker et al., 2024) is a training-free reconstruction based fake image detection technique. The first requirement is to collect an ensemble of VAE's of prominent latent diffusion models. Given an image, it is first reconstructed using the VAE and then the reconstruction is saved. The original image as well as the reconstruction are passed through the VGG16 (Simonyan & Zisserman, 2015) and the LPIPS distance is computed. The key hypothesis is that, a fake image can be reconstructed in an easier manner than a real image, therefore the distance will be lower. In the paper, the authors provide a plot showing the distribution of real and fake images. Based on this plot and our trials with the model, we pick a threshold of 0.018 for classification.

### A.2 ABLATIONS

### A.2.1 COMPUTATIONAL EFFICIENCY

Training a fake image detector requires generating a large number of images. This can become computationally heavy when using latent diffusion models, which utilize multiple rounds of forward pass through the UNet ($\epsilon_\theta$) and final decoding through the $\phi_{dec}$. Additionally, a text encoder is also used to condition the generation on text prompts. Since our approach neither utilizes the text encoder nor the UNet, and only generates images with a single forward pass through $\phi_{enc}$ and $\phi_{dec}$, it is much more efficient. We measure this difference in terms of the number of multiply-accumulate operations needed to generate the 179257 fake images. We ensure ours and the baseline are generating images at the same resolution.

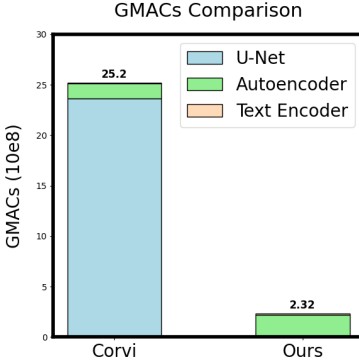

Figure 5: Computational cost measured in the number of multiply-accumulate operations. Ours is more than 10x efficient than the state-of-the-art method of (Corvi et al., 2022). Note that text encoder cost is relatively negligible compared to the U-Net and autoencoder.

**Results and Discussion**: Figure 5 shows the results, where our method of curating the dataset is ten times more cost-effective than the existing state-of-the-art approach. Unsurprisingly, majority of the cost comes from running the UNet. By skipping the UNet step, we are able to reduce the computational cost. Furthermore, our approach can maintain similar effectiveness even with lesser data, compared to the full dataset, setting it apart from Corvi. We discuss this in detail in Appendix 5.4.

### A.2.2 FALSE PATTERN LEARNING IN OJHA-LDM

Ojha et al. (2024) train a Universal Fake Image detector by linear probing a CLIP backbone on ProGAN (Karras et al., 2018) generated images. However, they also train a version trained on LAION (Schuhmann et al., 2022) and LDM (Rombach et al., 2022) images. This model (Ojha-LDM) ends up associating downsampling with real images. We demonstrate this using a simple

experiment. First evaluate Ojha-LDM on our set of real images and SD generated images. Now downsample both sets of images to 256 x 256 and measure the performance again.

|  | **Real** | **SD** |
|---|---|---|
| *Original* | 54.16 | 69.56 |
| *Downsampled to 256 x 256* | 83.58 | 29.83 |

Table 5: **Effects of resizing on Ojha-LDM**. Real accuracy improves when the same real images are downsampled. The fake accuracy on SD images also drops. This shows that images which have downsampling artifacts are likely to be identified as real images by Ojha-LDM.

Table 5 shows the results of our experiment. We can see that downsampling real images increases the chance of the detector identifying them. But at the same time, the fake accuracy decreases.

**Analysis**: Such false patterns are also caused by lack of a well-aligned real-vs-fake dataset. We look into the dataset that Ojha-LDM was trained on. Their fake images come from LDM, and are generated at a resolution of 256 x 256. However, their real images which come from LAION are present in a variety of resolutions. However, they were resized to 256 x 256 during training. During training, the model is able to use some of these artifacts to fit the training distribution. This further suggests that without proper dataset alignment, the detector can very easily pick up on spurious features present in the data.

### A.2.3 THRESHOLD-LESS EVALUATION

In Tables 1, 2, and 4, we assess the accuracy of various detectors under a fixed threshold, simulating a test environment. In this section, we present the results for two important evaluation metrics: average precision (AP) and true positive rate at a 5% false positive rate (tpr@5fpr). These metrics serve as indicators of the classifier's maximum ability to correctly identify fake images. For our evaluation, we use the same test dataset described in Section 5.3. Specifically, we examine both the original images, as referenced in Table 1, and the post-processed images, which are presented in Table 2, grouping them into the same category for a comprehensive analysis. The dataset consists of 6000 real images and 6000 images for each of the respective categories.

The results for average precision (AP) and true positive rate at 5% false positive rate (tpr@5fpr) are reported in Tables 6 and 7, respectively. From these tables, we observe that classifiers trained on a well-aligned dataset demonstrate near-perfect separability, as evidenced by the high AP scores and the tpr@5fpr values. Furthermore, the linear-probing based methods (Ojha et al., 2024; Cozzolino et al., 2024) as well as the training-free, reconstruction based method (Ricker et al., 2024) achieve a low AP, tpr@5fpr. This shows that these approaches cannot be improved by calibrating the threshold.

### A.2.4 IMPACT OF ARCHITECTURE CHOICE AND INITIALIZATION

In this section, we experiment with different architectural choices and initializations in order to disentangle the effects that a particular architecture/initialization may have on the results. We experiment with the following variation,

- **Same Architecture, Different Initialization:** We use the same ResNet-50, but we use the DINO initialization (Caron et al., 2021). DINO is a self-supervised, self-distillation based approach to pre-train image feature extractors. We also fine-tune a FCN (Long et al., 2015) backbone which was originally trained on the semantic segmentation task on the (Lin et al., 2015) dataset.
- **Different Architecture, Same Initialization:** We fine-tune a Wide-ResNet (Zagoruyko & Komodakis, 2017) pre-trained on ImageNet. Wide-ResNets are widened versions of original ResNet with decreased depth. We also perform the same experiment with the ViT-B/16 (Dosovitskiy et al., 2021) architecture. For ViT-experiments, we crop the image to 224x224 due to the input resolution requirements.
- **Different Architecture, Different Initialization:** We fine-tune a modified ResNet trained using the CLIP (Radford et al., 2021) objective. The CLIP ResNet is deeper and bigger

| Method | SD | MJ | Kandinsky | Playground | PixelArt-$\alpha$ | LCM |
|---|---|---|---|---|---|---|
| AEROBLADE (Ricker et al., 2024) | 90.81 | 96.48 | 94.03 | 71.53 | 87.84 | 89.99 |
| Ojha-ProGAN (Ojha et al., 2024) | 62.02 | 52.33 | 64.55 | 61.58 | 61.45 | 64.75 |
| Ojha-LDM (Ojha et al., 2024) | 61.93 | 61.72 | 74.21 | 69.39 | 68.72 | 70.05 |
| Cozzolino-LDM (Cozzolino et al., 2024) | 71.63 | 73.72 | 74.16 | 74.52 | 75.90 | 70.95 |
| Corvi (Corvi et al., 2022) | 97.87 | 94.81 | 95.32 | 90.93 | 94.16 | 96.15 |
| *Ours* | 99.32 | **98.37** | 97.97 | 98.17 | 98.38 | **99.84** |
| *Ours-Sync* | **99.40** | 98.30 | **98.14** | **98.54** | **98.58** | 99.79 |

Table 6: **Average Precision**. AP of different methods for detecting real and fake images. Our approach shows better separability between real and fake images across various settings as indicated by the AP. We

| | SD | MJ | Kandinsky | Playground | PixelArt-$\alpha$ | LCM |
|---|---|---|---|---|---|---|
| AEROBLADE (Ricker et al., 2024) | 59.08 | 85.31 | 69.66 | 13.57 | 50.95 | 54.62 |
| Ojha-ProGAN (Ojha et al., 2024) | 13.28 | 9.33 | 17.08 | 12.20 | 12.85 | 16.27 |
| Ojha-LDM (Ojha et al., 2024) | 12.63 | 14.32 | 26.34 | 16.58 | 17.56 | 18.35 |
| Cozzolino-LDM (Cozzolino et al., 2024) | 18.66 | 21.46 | 21.70 | 20.21 | 24.10 | 20.76 |
| Corvi (Corvi et al., 2022) | 91.69 | 81.46 | 83.86 | 65.75 | 83.66 | 84.32 |
| *Ours* | 96.82 | **92.19** | 91.00 | 91.52 | 94.25 | **99.02** |
| *Ours-Sync* | **97.08** | 91.77 | **91.08** | **92.88** | **94.58** | 98.53 |

Table 7: **True Positive Rate (TPR) at 5% False Positive Rate (FPR)** for different methods across various evaluation settings. The best results for each setting are highlighted in bold.

than the ResNet that we used for our earlier experiments. In order to process images of varying scales, we replace the attention pooling with adaptive average pooling.

For our new variants, we follow the practice of Gragnaniello et al. (2021) by modifying the stem where we remove the downsampling operations. For each configuration, we train two variants, one of them is trained on the dataset used by Corvi et al. (2022) and the other one is trained on our dataset (Ours-Sync). We measure performance using the AP metric that we used in Appendix A.2.3. We report the results in Table 8.

We observe that the detectors trained using an aligned dataset exhibit superior performance to their counterparts irrespective of the network architecture and initialization. Furthermore, among the detectors trained using the dataset used by Corvi et al. (2022), we observe that the results are mostly similar, except for the CLIP-initialized Modified ResNet. The CLIP ResNet has more parameters in comparison to the other networks, therefore we hypothesize that it might have overfit more to the spurious correlations present in the training data. ViT-based detectors perform worse than CNN-based detectors, we hypothesize that this is an effect of patch-based training in CNN-architectures which show better generalization.

### A.2.5 ROBUSTNESS TO POST-PROCESSING

Previously, we evaluated our model on several post-processing operations. We also examined the sensitivity of our approach to resizing (Fig 2) and .webp compression. In this section, we study in-detail the robustness of our method to additional post-processing operations which are commonly found in the real world. We experiment with the following post-processing operations,

- **Blur:** We blur the images by using a gaussian kernel of size 9. We vary the standard deviation of the kernel, denoted by $\sigma$.

- **Gaussian Noise:** We add gaussian noise to original image. We control the standard deviation of the added noise in order to study the behaviour of our model.

- **JPEG Compression:** We study the effects of JPEG compression by varying the compression quality.

| Initialization | Architecture | Dataset | SD | MJ | Kandinsky | Playground | PixelArt | LCM |
|---|---|---|---|---|---|---|---|---|
| ImageNet | ResNet-50 | Corvi | 97.87 | 94.81 | 95.32 | 90.93 | 94.16 | 96.15 |
| | | Ours | 99.40 | 98.3 | 98.14 | 98.54 | 98.58 | 99.79 |
| DINO | ResNet-50 | Corvi | 98.27 | 95.26 | 96.25 | 94.26 | 95.09 | 97.05 |
| | | Ours | **99.52** | **98.52** | **98.81** | **98.96** | 99.09 | **99.84** |
| FCN | ResNet-50 | Corvi | 98.29 | 95.81 | 96.14 | 93.45 | 94.97 | 96.23 |
| | | Ours | 99.27 | 98.4 | 97.86 | 98.88 | **99.12** | 99.83 |
| ImageNet | Wide-ResNet | Corvi | 97.72 | 94.82 | 95.59 | 91.54 | 94.54 | 95.31 |
| | | Ours | 99.06 | 98.22 | 97.23 | 97.12 | 98.13 | 99.51 |
| ImageNet | ViT-B/16 | Corvi | 85.78 | 66.2 | 72.57 | 65.42 | 65.61 | 71.98 |
| | | Ours | 93.68 | 89.46 | 87.85 | 82.01 | 92.58 | 91.58 |
| CLIP | Modified-ResNet | Corvi | 96.90 | 93.86 | 93.39 | 88.09 | 93.6 | 91.02 |
| | | Ours | 99.12 | 98.44 | 97.17 | 96.86 | 97.99 | 99.36 |

Table 8: **Average Precision (AP) Across Architectures and Initializations.** Performance of networks with different initializations and architectures, trained on the dataset provided by Corvi and our dataset. A detector trained on a well-aligned dataset consistently outperforms the Corvi baseline. This indicates that the positive effects of dataset alignment are independent of the choice of architecture and pretraining.

Our design of the perturbations follows AEROBLADE (Ricker et al., 2024). For these experiments, we use the same images from our experiments in Section 5.2. Sensitivity of our detector to blur, noise and compression can be found in Fig 6, 7 and 8 respectively.

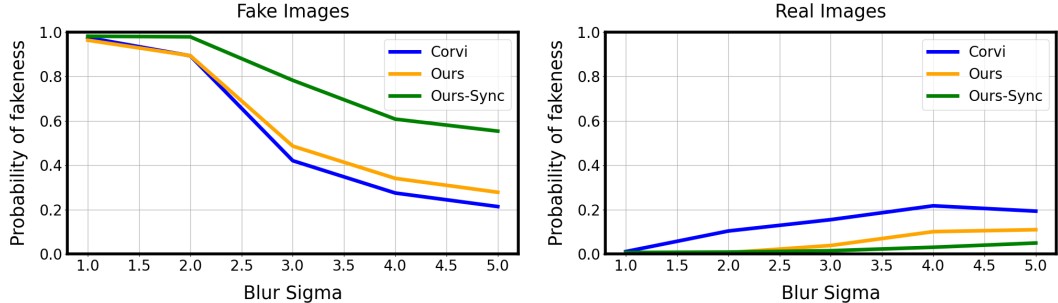

Figure 6: **Sensitivity of fake detectors to image blurring** for a set fake images (left) and a set of real images (right). We use a kernel size of 9 and vary the standard deviation. Ours-Sync shows increased robustness to blurring showing the importance of batch-level alignment.

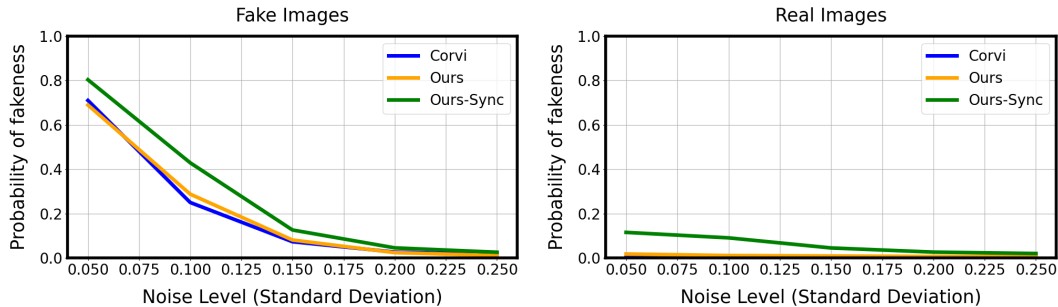

Figure 7: **Sensitivity of fake detectors to additive noise** for a set fake images (left) and a set of real images (right). We control the noise level by varying the standard deviation of the added noise.

There is a huge performance gap between the Ours-Sync method and the others when the image is blurred. This adds to our earlier observations from Tables 1 and 2 that ensuring batch-level alignment is a very important design choice along with the design of the dataset. Furthermore, the general robustness observed also shows that our detectors are not just looking at some low-level traces which can be washed away by simple post-processing operations.

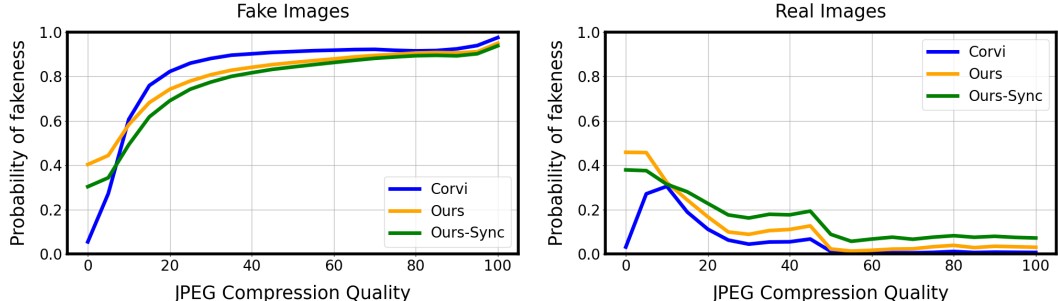

Figure 8: **Sensitivity of fake detectors to JPEG Compression** for a set fake images (left) and a set of real images (right). We control the compression level by varying the JPEG compression quality.

