# OpenReview forum: "Aligned Datasets Improve Detection of Latent Diffusion-Generated Images"
_ICLR.cc/2025/Conference — ICLR 2025 Poster_

### Official Review · Reviewer_ecmv · 2024-10-20

**Soundness:** 2
**Presentation:** 3
**Contribution:** 3
**Rating:** 6
**Confidence:** 4

**Summary:**

The paper mainly focuses on detecting the fake images generated by latent diffusion models (LDMs). It states that dataset alignment plays an important part in detecting LDM-based fake images and proposes a simple method to detect LDM-based fake images by training detectors with the aligned dataset (real images and fake images created by only LDM's autoencoder). Experiments show that the proposed method is inexpensive and effective in detecting LDM-based fake images.

**Strengths:**

1. The proposed method is simple and can potentially be a useful baseline for LDM-based fake image detection.
2. The proposed method is more effective and efficient than the paper's major baseline (Corvi).
3. Extensive experiments provide sufficient support for the superior performance of the proposed method. 4. The paper is somewhat clear and well-written to me.

**Weaknesses:**

1. The title slightly disqualifies the paper's main topic since the authors pay attention to LDM-based fake images. However, many generative models can create general fake images, including AutoEncoder, GAN, VAE, and LDM. It seems more appropriate to add qualifiers in the title.
2. Both Corvi and the proposed method remain vulnerable to minor changes. Although the proposed method shows wonderful performance against some LDM-based images, the comparison between results from Table 1 and Table 2 reveals both Corvi and the proposed method can be easily disturbed by post-processing like lossy compression.
3. Lack of further insights between "Ours" and "Ours-Sync". In section 4 (Our Approach), two variants "Our" and "Our-sync" are introduced but with no extra explanations. In experiments, "Ours-Sync" shows superior performance in most cases than "Ours" with almost no analysis.
4. The figure in the conclusion loses its caption. Help it.

**Questions:**

1. Why does the performance of detectors drop a lot facing fake images with small scaling factors or real images with large factors? Are there any reasonable explanations?
2. Why does "Ours-Sync" outperform "Ours" in most cases, and why does "Ours" kick back in some special cases?
3. The combination of the proposed method and Corvi should be reasonable and available. Does it perform remarkablely in detecting LDM-based fake image detection?
4. It seems that the proposed method can enjoy a better performance with a large batch size. Would you mind a try?
5. I understand that using ResNet-50 is for alignment in the network architecture to some baselines, but more experiments on different network architectures are suggested.

---

> ### Author Response · Authors · 2024-11-19
> **Authors' Response**
>
> 1. **Title slightly disqualifies the paper’s main topic**\
> We thank the reviewer for raising this point. We will add qualifiers to the title which represents problem statements (only Latent Diffusion). We will do so in the final version of the paper.
> 2. **Both Corvi and the proposed method remain vulnerable to minor changes**\
> While we agree that both Corvi and our proposed method can be disturbed by post-processing, the degree to which they get disturbed is very different. For example, if we look at Figure 2 and consider 0.5 as the threshold for real to fake prediction transition, we see that the Corvi baseline starts calling an image fake even if one simply resizes it to 0.75 its size, whereas to do the same to our detector, one will need to downsize it to more than half its size (at 0.45). And we see a similar story for real images (lines 330,331). So, even though the problem of reliably detecting fake images is still not completely solved, our proposed method is still much more robust compared to the existing paradigm, while at the same time being computationally more efficient.
> 3. **Lack of further insights between Ours and Ours-Sync**\
> We thank the reviewer for asking for a clarification on this matter. Since our goal is to force the detector to focus only on the autoencoder's artifacts while classifying an image as fake (and real when those artifacts are absent), the real and fake datasets need to be aligned, so that all other semantic differences between them are minimal except for the autoencoder's artifacts, i.e., for every x in the real set, there is a x' in the fake set, where x' is the autoencoder reconstruction of x. However, if x and x' are in separate batches, the network can learn to associate x->real and x'->fake independently of each other, and hence might use some semantic information to learn that mapping. If, on the other hand, x and x' are part of the same batch, in each update of its weights, the model is learning to not look at the semantics. This is what we refer to as batch-level alignment (line 239). We will include a more thorough discussion about the difference between the two modes of training in the final version.
> 4. **The figure in page 10 loses its caption**\
> We have properly numbered the figure and referenced it in the updated version of main text along with a proper caption.
>
> 5. **Ensemble of Corvi and Ours method**\
> We have created a fusion version of Corvi and Ours-Sync and evaluated it. We do so in two ways, fusion-max and fusion-avg. In fusion-max, we classify an image as fake if either detector classifies it as fake. In fusion-avg, we assign equal weightage to both models and take the average prediction of both the networks and classify the image based on that. We evaluate the models in terms of average precision (AP). We calculate the AP by combining the original and post-processed version of each evaluation setting (SD, MJ etc). This way, we get a single score which represents the setting.
>
> | Average Precision   | SD   | MJ   | Kandinsky   | Playground   | PixelArt-α   | LCM   |
> |------------|------------|------------|------------|------------|------------|------------|
> |  Corvi    | 97.87       | 94.81       | 95.32       | 90.93       | 94.16       | 96.15      |
> | Ours-Sync      | **99.4**      | **98.3**       | **98.12**       | **98.54**       | **98.58**  | **99.79**       |
> | fusion-avg     | 98.55       | 96.12       | 96.2       | 94.55       | 95.75       | 97.97       |
> | fusion-max      | 99.4       | 98.11      | 97.97       | 98.29       | 98.5       | 99.67       |
>
> We observe a performance degradation in the fusion models compared to the Ours-Sync model. We hypothesize that this is due to spurious correlations learned by the Corvi method, which weaken the Ours-Sync model's predictions in some cases, primarily in fusion-avg, where Corvi’s prediction always impacts the ensemble network's overall prediction.
>
> 6. **More Architectures**\
> We include 2 more architectures (with different pre-training configurations) in order to demonstrate our point regarding dataset alignment. We modify the Wide-ResNet [1] by removing upsampling operations from its stem, following [2], and use it to train a fake image detector. Additionally, we use a larger, modified ResNet pretrained with the CLIP objective. For each setup, we compare detectors trained on the Corvi baseline dataset against those trained on our dataset.The results can be found in **Appendix A.2.4**. The detector trained on reconstructions consistently outperforms the Corvi baseline. Our experiments show that the effectiveness of data alignment is  independent of architecture choices/initializations.
> **References**\
> [1] Zagoruyko, Sergey. "Wide residual networks." arXiv preprint arXiv:1605.07146 (2016).\
> [2] Gragnaniello, Diego, et al. "Are GAN generated images easy to detect? A critical analysis of the state-of-the-art." 2021 IEEE international conference on multimedia and expo (ICME). IEEE, 2021.

---

> ### Author Response · Authors · 2024-11-19
> **Authors' Response (continued)**
>
> 7. **Why does the performance of detectors drop a lot facing fake images with small scaling factors or real images with large factors? Are there any reasonable explanations?**\
> We think the reviewer is referring to Figure 2 of the main paper, where we see that, for our method as well as for Corvi, the probability of an image being fake drops when the downscaling factor reaches x4 (at 0.25). We think that this is because the downscaling of x4 was never seen during training. The maximum resolution of our training images are 640 x 640, which can, in very small fraction of cases, can get resized to 256 x 256, i.e., a downsizing factor of x2.5. And since CNNs are not scale invariant, the learnt filters won't behave the same for smaller images. This reason will be applicable to both our method as well as Corvi, since the training real data, as well as the data processing pipeline is the same for both. As for real images, Corvi again struggles with high scaling factors (probability of fakeness increases). Our method, on the other hand, is much more robust in accurately  classifying them (a low score) for all scaling factors. We will include a discussion in the final version for why both methods struggle when fake images are downsized by large factors.
> 7. **Larger Batch Size**\
> We thank the reviewer for the interesting suggestion. We conducted an experiment where we set the batch size to be 1024: 512 real and 512 fake images (our main paper had 64 real and 64 fake images during training).
>
> | Average Precision   | SD   | MJ   | Kandinsky   | Playground   | PixelArt-α   | LCM   |
> |------------|------------|------------|------------|------------|------------|------------|
> |  Ours-Sync      | 99.4      | 98.3       | 98.12       | **98.54**      | 98.58  | 99.79       |
> | Ours-Sync (1024 batch size)     | **99.44**      | **98.5**       | **98.46**       | 98.47       | **98.64**  | **99.81**       |
>
> We observe a slight performance improvement with an increased batch size. This may be attributed to the use of well-aligned datasets, where the gradient updates after each batch are relatively less-noisy. As a result, the training dynamics do not significantly benefit from a larger batch size.

---

> > ### Author Response · Authors · 2024-11-23
> > **Authors' Response**
> >
> > Please let us know if our rebuttal addresses your concerns, and if we could provide further clarifications that would strengthen your approval of the paper.

---

> > > ### Comment · Reviewer_ecmv · 2024-12-02
> > >
> > > Thanks for the detailed response from the authors, leading to an excellent illustration of the manuscript. Therefore, I keep my final rating and commend their work to the conference.

---

> > > > ### Author Response · Authors · 2024-12-02
> > > > **Authors' Response**
> > > >
> > > > We thank the reviewer for stating that our paper is well-illustrated and recommending its acceptance.

---

### Official Review · Reviewer_caUt · 2024-10-30

**Soundness:** 2
**Presentation:** 1
**Contribution:** 3
**Rating:** 6
**Confidence:** 4

**Summary:**

This paper investigates the effectiveness of dataset alignment in fake image detection, specifically for images generated by latent diffusion models (LDMs). The authors propose a simple approach: reconstructing real images using the LDM's autoencoder and treating these reconstructions as fake images. This aligned dataset allows for training a classifier to distinguish real from fake images, focusing on the generative model's artifacts rather than spurious correlations such as resolution or content differences. Experimental results show that this alignment method improves the robustness and generalization of the detector across different generative models, while also being more computationally efficient.

**Strengths:**

1. The paper proposes an interesting framework for generating fake images by using only the LDM's autoencoder, rather than the full generative process, to train the detector. Experiments show this approach achieves better performance compared to using complete generated images, which is a noteworthy finding.

2. The paper reveals the impact of scaling factors on detection methods, contributing valuable insights to the field.

**Weaknesses:**

1. Lack of technical contribution: Although the findings are interesting, the authors did not deeply explore the reasons behind the observed phenomena or how they could further benefit existing detection methods. They only conducted shallow comparative experiments using their generated data. Overall, the contribution feels insufficient.

2. Limited comparison methods: The authors aim to prove their dataset has better generalization and performance than others, but they only compared it with two methods. These methods are not state-of-the-art (SOTA) on complex public datasets like GenImage. The lack of sufficient comparisons weakens the paper's persuasiveness. Additionally, the comparison datasets are not from AIGC detection benchmarks, making it difficult to determine a fair performance improvement.

3. The proposed method of classifying only LDM-generated data can avoid some spurious features but also ignores certain important features, such as artifacts introduced during the diffusion process. The method is mainly effective for LDM-based generation techniques and less so for traditional GAN or diffusion models. It may be that the LDM itself removes diffusion artifacts, making it easier to detect LDM-specific artifacts. Moreover, the method cannot handle semantic inconsistencies left by some generative methods, such as incorrectly generated hands.

4. The authors should conduct robustness tests, such as robustness against noise or JPEG compression, to further validate the method.

5. The paper structure could be improved, with clearer diagrams and better formatting. The dataset descriptions are somewhat disorganized, and a more structured introduction is recommended.

Despite these shortcomings, I still believe this discovery is valuable, and the relative improvements are significant. I encourage the authors to further explore this phenomenon in depth.

**Questions:**

See the weakness section.

---

> ### Author Response · Authors · 2024-11-19
> **Authors' Response**
>
> 1. **Authors did not deeply explore reasons behind phenomena and have conducted shallow comparative studies**\
> We respectfully disagree with the reviewer regarding the lack of depth in the exploration of the phenomena observed. We have identified two major kinds of spurious correlations relating to resizing and compression and explained precise reasons for both of them in **Section 5.2** and the **Discussion** section respectively. We also identified and explained a similar spurious correlation which can occur even when using frozen feature extractors such as CLIP (**Appendix A.2.2**).
> \
> \
> Furthermore, we also show the effectiveness of dataset alignment in terms of **mitigating spurious correlations (Sec 5.2)**, **computational efficiency (Sec 5.3)**, **general performance (Sec 5.4)**, **semantics-independent pattern learning (Sec 5.5)**, and **Data efficiency (Appendix A.2.1)**. By analyzing the approach along these different dimensions, we have identified different desirable properties which the approach brings. It would be helpful if the reviewer could give suggestions regarding the nature of the experiments that would make the work stronger.
> 2. **Limited Comparison methods**\
> The reviewer mentions that we only compared it with 2 methods, however we wish to highlight that **we have compared our models with 5 baseline methods** (details listed in Sec 5.1), these baselines were carefully selected in order to represent 3 different paradigms of fake image detection methods-training based (Corvi), linear-probing based (Cozzolino-LDM, Ojha-ProGAN and Ojha-LDM) and reconstruction based (AEROBLADE).
> 3. **Lack of results on public benchmarks**\
> Our work focuses on training a reliable fake image detector for a family of latent diffusion models. Therefore, in order to make sure that it is clearly the case, we wanted to evaluate across a wide variety of latent diffusion models. To the best of our knowledge, existing benchmarks such as GenImage possess a limited number of Latent Diffusion models and are tailored towards Universal Fake Image detection. For instance, they do not contain images from the latest models such as Playground, PixelArt, Kandinsky and Latent Consistency Models. Therefore, we decided to create our own evaluation benchmark, specifically for the problem that we were interested in solving - that is, reliably detecting fake images from the LDM family.
> \
> \
> However, we have also tested our model on the Sentry benchmark [1]. The Sentry benchmark consists of  images from 2 latent diffusion models, SDv1.5, SDv2 and the closed-source commercial model Midjourney. We report our results in terms of AP and accuracy with a fixed threshold (0.018 for AEROBLADE, 0.5 for everything else). In order to calculate the AP, we use the same real images which we use in the rest of the paper. With SDv1.5, the authors also generate images which “look realistic”, which they label as SDv1.5R.
> The results indicate that our approach’s performance is similar to what we reported. Our approach achieves near perfect performance in terms of accuracy and AP on the SDv1.5, SDv1.5R and SDv2 images. It is important to note that the Midjourney (Mjv5) images used in the Sentry benchmark are gathered from the internet based on community uploads. It simulates the post-processing setting that we report in our paper (Table 2). On Midjourney, our detector achieves a good AP value of 94.37%. These results show that the improvements that we observe through the use of data alignment is not tied to our curated dataset.
>
> | Accuracy   | SDv1.5   | SDv1.5R   | SDv2   | Mjv5   |
> |------------|------------|------------|------------|------------|
> | AEROBLADE      | 99.23       |    93.94    | **99.75**       | 29.16      |
> | Ojha-ProGAN      | 18.08       | 2.43       | 3.54       | 4.64       |
> | Ojha-LDM      | 82.63       | 27.39       | 29.42      | 49.29       |
> | Cozzolino-LDM      | 76.41       | 48.3       | 58       | **66.08**       |
> | Corvi      | **100**       | 99.97       | **99.72**       | 12.42       |
> | Ours      | **100**       | **99.98**       | 97.88       | 49.13       |
> | Ours-Sync      | 99.99| 99.91 | 99.01 | 65.35      |
>
> | Average Precision   | SDv1.5   | SDv1.5R   | SDv2   | Mjv5   |
> |------------|------------|------------|------------|------------|
> | AEROBLADE      | 99.42       |    99.04    | 99.51       | 79.06      |
> | Ojha-ProGAN      | 91.27       | 79.71       | 81.91       | 58.32       |
> | Ojha-LDM      | 94.42       | 78.68       | 79.29      | 69.71       |
> | Cozzolino-LDM      | 94.05       | 86.23       | 79.93       | 78.79       |
> | Corvi      | **100**       | **99.99**       | **99.96**       | 60.86       |
> | Ours      | **100**       | **99.99**       | 99.68       | 91.07       |
> | Ours-Sync      | 99.99| **99.99** | 99.83 | **94.37**      |
>
> **References**\
> [1] Lu, Zeyu, et al. "Seeing is not always believing: benchmarking human and model perception of AI-generated images." NeurIPS 36 (2024).

---

> > ### Author Response · Authors · 2024-11-19
> > **Authors' Response (continued)**
> >
> > 4. **Method is specific to LDM’s, cannot detect images from GAN’s etc**\
> > We would like to clarify that we are not trying to solve the problem of universal fake image detection in this work. While the community emphasizes on solving the problem of universal fake image detection, in this work, we highlight that reliable detection of fake images from a single family of models is not by any means a trivial task and highlight it by showing several failure modes for a lot of existing models such as spurious correlations (**Sec 5.2, Appendix A.2.2**) and limited performance (Tab 1, 2 and Discussion section). We believe that before tackling the problem of Universal Detection, it is important to build reliable detectors for a single class of models. Please refer to our general comment addressed to all reviewers which elaborates on this point.
> > 5. **Authors should conduct Robustness test**\
> > We thank the reviewer for the suggestion and have tested the sensitivity of our method to blur, additive noise and JPEG compression. The results can be found in **Appendix A.2.5** of the updated submission. We observe that our method demonstrates increased robustness to blurring compared to the Corvi baseline. Furthermore, the general robustness to compression, blurring, and noise indicates that our detector is not reliant on low-level signals, which can be eliminated by simple post-processing operations.
> > 6. **The paper structure could be improved**\
> > First, we would like to point out that multiple reviewers (hbmp, ecmv) did find the paper well written. Nevertheless, we will be more than happy to improve the paper in any respect, whether it is writing or figures. However, we do not know what exactly the reviewer intended. It will be helpful if the reviewer can specifically list out the sections which need improvement and in what way; and how exactly we can improve the diagrams. We will do our best to incorporate those comments in our final version.

---

> > > ### Comment · Reviewer_caUt · 2024-11-23
> > >
> > > Thank you for your response and additional experiments. While I have carefully reviewed your rebuttal and manuscript, and appreciate that you have addressed some of my concerns, I still have several remaining reservations:
> > >
> > > 1. Your core finding essentially relies on training the detector to learn only the low-level artifacts left by the autoencoder while maintaining alignment with real images in other aspects. However, this approach risks missing important high-level semantic information, such as commonly occurring text errors or additional fingers in generated images. These high-level artifacts could be overlooked by your method.
> > >
> > > 2. Regarding the resizing issue: Traditional detection models typically resize images to 224 or 256 pixels primarily due to inference and training computational constraints. While achieving good results without resizing is straightforward, I believe you've omitted an important baseline: comparing your method against training with fake images resized to match real image dimensions. Additionally, your non-resizing approach would significantly increase both inference and training times, which is a notable disadvantage.
> > >
> > > 3. Concerning my previous Question 1: I wanted to explore why using just an autoencoder can achieve such good results - surely it's not solely due to resolution differences? The original Corvi method must have also captured autoencoder artifacts, so why was its generalization not as strong? I would have preferred to see a more in-depth analysis of these aspects.
> > >
> > > 4. While your response regarding mitigating spurious correlations (Sec 5.2) is informative, I find it insufficient. The computational efficiency analysis is rather obvious and could be moved to supplementary materials. In real-world scenarios, dataset creation efficiency is less critical than inference time and computational costs - areas where your method may be disadvantaged due to resolution requirements.
> > >
> > > 5. Your comparison methods appear too basic, focusing on simple detection techniques. It's unclear how your approach competes with current SOTA AIGC detection methods like DIRE and NPR. However, I do find your relative improvements over the Corvi method and the analysis in supplementary section A.2.4 quite valuable.
> > >
> > > 6. Section A.2.4 is particularly valuable and should be incorporated into the main text. Consider adding ViT-based methods, as the current focus is exclusively on CNN architectures.
> > >
> > > In conclusion, while there are several concerns, your findings are intriguing.

---

> > > > ### Author Response · Authors · 2024-11-24
> > > > **Authors' Response**
> > > >
> > > > We address the concerns raised by the reviewer below, some of which are new and were not mentioned in the original review
> > > >
> > > > **1.1) Approach could miss out on high-level semantic information**\
> > > > The reviewer is correct that we are indeed pushing the detector to focus less on the typical high level features (e.g., hand with 6 fingers) and more on just the artifacts left only by the autoencoder. However, the resulting detector can still principally detect all kinds of fake images as fake, including those which have semantic anomalies like distorted text and additional fingers.This is because even those kinds of images do pass through the decoder, and have its artifacts imprinted in them (lines 94-96). This is shown empirically as well, where we train our detector on aligned real/fake dataset, but test it on fake images which are generated using text (e.g., SD, MJ in Table2). Images from these categories of fake images could indeed have semantic anomalies, but our method does well in detecting them as fake, as seen in Tables 1 and 2.
> > > > \
> > > > **1.2) Approach relies only on low-level artifacts left by the autoencoder.**\
> > > > As mentioned above, while we do indeed want to align the real and fake images as much as possible, we want to clarify that our detector is not solely looking for extremely low level details. This is because if it did, it would have been very easy to make the detector ineffective by simple resizing or blurring operations. But our method is particularly robust to these operations (e.g., resizing, blurring, JPEG compression and gaussian noise); please see **Fig. 2, Table 2**, and **Fig. 6**. Finally, another result which hints that it is not solely looking at the low level details is the difference in performance of our detector when we train it on  natural real/fake images vs when we train it on the Shaders dataset. The results in **Table 3** show that our detector trained on natural images performs better than our detector trained on Shaders images, thereby indicating that our method is capable of detecting semantic features when necessary.
> > > > \
> > > > **2.1) An important baseline is missing: comparing your method against training with fake images resized to match real image dimensions.**\
> > > > Even if one resizes the fake images to match the resolution of real images at the dataset creation stage, they would still introduce the same upsampling artifacts, which would be present in fake images but not in real images. This is just the same problem that Corvi method had, where the upsampling artifacts created as part of the data processing pipeline are mainly present in fake images (please see a detailed explanation in **lines 290-302**). So, no, we don’t think this baseline that the reviewer is suggesting is any different than the Corvi method, because the difference is only in when/where the asymmetric upsampling is happening; but it will happen for both. Finally, we have actually considered a baseline which indeed does the thing that the reviewer is suggesting. In **Appendix A.2.2**, we show that the Ojha-LDM method learns the exact same spurious correlation (all real images are downsized to meet the resolution of the fake image).\
> > > > **2.2) Additionally, your non-resizing approach would significantly increase both inference and training times, which is a notable disadvantage.**\
> > > > We feel there are many points of confusion in this statement by the reviewer. First, we want to clarify and reiterate that we use the exact same data processing pipeline as Corvi. During training this includes a sequence of ‘random_crop’ and then ‘resize’ to 256 x 256 resolution (**lines 290-302**), of which we finally take a 96 x  96 crop before feeding it into the network (**line 708**). So, for both ours as well as Corvi, the network is trained only on 96 x 96 crops. The only difference is that the original fake images on which these operations as performed are different; they are 256 x 256 for Corvi, and in our case are autoencoder reconstructions of real images. The network is trained on the same resolution images (96 x 96) and will have exactly the same computational cost for ours vs Corvi’s method. Second, during testing time, we again keep the same data processing pipeline as the Corvi method. Neither our method nor Corvi perform any resizing operation to the test images; since the detector has a ResNet backbone, we process the original raw input without needing to resize. We will add this detail in the appendix in the final version. In summary, there is no computational disadvantage of our method compared to Corvi. In fact, our dataset creation process is significantly faster compared to Corvi (Fig. 3), something that the reviewer also acknowledges in their 4th point in this comment.

---

> > > > > ### Author Response · Authors · 2024-11-24
> > > > > **Authors' Response (continued)**
> > > > >
> > > > > **2.3) Achieving good results without resizing is straightforward**\
> > > > > If the reviewer is suggesting that the only reason our method is working is because we have not done resizing, then that is not correct. As mentioned in our previous response, **we do in fact perform the resizing operation, same as Corvi**. The difference is that we do it on real/fake images that are aligned. If, on the other hand, the reviewer was suggesting that Corvi method could be just as good if they had not performed any resizing, through random_resized_crop operation explained in lines 293-302, then again that is not accurate. This is because the function of random_resized_crop is critical for the Corvi method in making sure the detector gets to see many variants of the same image real/fake image (through different sizes of random crops) thereby learning invariance to scale (CNN’s are not fundamentally scale invariant). So, we don’t agree that simply not doing the resizing operation can solve the problem.\
> > > > > **3) The original Corvi method must have also captured autoencoder artifacts, so why was its generalization not as strong?**\
> > > > > To our understanding, the original Corvi could have latched onto the artifacts due to: (i) the diffusion UNet, (ii) autoencoder, (iii) resizing artifacts. We have shown empirically that it was at least looking at the resizing artifacts (iii) in Fig. 2. And the reviewer is right; that detector could also have been looking at the autoencoder’s artifacts. How important each of these three factors (there could be others as well) is not clear. The reason we believe that our method is effective is because all the images generated by an LDM will have the autoencoder’s artifacts (ii), even those with semantic anomalies like 6 fingers. Hence, we tried to limit its focus just on the autoencoder’s artifacts because of which it is more robust to changes in resolution (Fig. 2). It is also more robust to changes in UNet architecture, which is supported by Corvi’s weak performance on Playground images (**Table 1**) which uses a UNet which is vastly different from LDM (**lines 423-426**). \
> > > > > **4) Response regarding mitigating spurious correlations insufficient. The proposed method may be disadvantaged due to resolution requirements.**\
> > > > > As we discussed in our response 2.2, our inference pipeline is exactly the same as Corvi, where both methods do no resizing operation on images at test time. More importantly, as seen by Fig. 2, if we wanted to gain some computational efficiency by downsizing both real and fake images, we can actually do that. E.g., even after resizing to half the size (0.5 on the x-axis), our detector is still able to call fake images as fake, and real images as real. This is not the case for the Corvi method. So, not only do we not have any computational issues at test time compared to Corvi, but we can increase our efficiency by downsizing without affecting the accuracy by much, something that might not be the case for Corvi. \

---

> > > > > ### Comment · Reviewer_caUt · 2024-11-26
> > > > >
> > > > > I previously had a misunderstanding about the resolution. I thought the authors hadn't performed the cropping step. Anyway, if the authors did not increase the inference resolution, then I believe this method has practical value and offers good insight. Thus I'm increasing my score and recommend acceptance. Additionally, regarding the comparative experiments, I still recommend that the authors replicate their dataset using some newer and more diverse backbones to further verify the method's generalizability in the final version.
> > > > >
> > > > > I also appreciate the authors' thorough response.

---

> > > > > > ### Author Response · Authors · 2024-11-28
> > > > > > **Authors' Response**
> > > > > >
> > > > > > We thank the reviewer for their constructive feedback, noting that the paper is insightful, holds practical value, and recommending its acceptance. In the final version, we will ensure the inclusion of results from more diverse backbones, such as ViTs.

---

> ### Author Response · Authors · 2024-11-24
> **Authors' Response (continued)**
>
> **5) Comparison techniques are too basic, compare against SOTA methods such as DIRE and NPR**\
> We respectfully disagree with the suggestion that the comparison techniques are too basic. In this response, we will show later on that the baselines we selected outperform the newly suggested baselines. Initially, the reviewer mentioned that we had compared our method to only two baselines, which was inaccurate. In fact, we compared our approach to five different baseline detectors. The reviewer’s new request is to compare two new methods which were not mentioned in the original review. And while we have still managed to get the results on them, we want to briefly mention some information about those methods. The reason we did not include the DIRE [2] method in our earlier draft was due to analysis that can be found in the AEROBLADE paper (Section 9 of the AEROBLADE paper). The paper experimentally analyzes spurious correlations learnt by the DIRE method (associating JPEG with real images and PNG with fake images). Nevertheless, we compare against both of these methods which the reviewer has now requested.
>
> For the evaluation, for completeness we use 2 checkpoints for NPR [1] as well as DIRE. For NPR, the authors provide checkpoints for the AIGCDetect [3] benchmark as well as the GenImage benchmark [4]. We report the performances of both of these checkpoints. Furthermore, we also use 2 of the checkpoints provided by the authors of DIRE, one which was trained on ADM-generated images [5] and one that was trained on Stable Diffusion v2 generated images. In order to give the complete picture, we report the AP (same setting as Appendix A.2.3).
> | **AP**            | **SD**    | **MJ**    | **Kandinsky** | **Playground** | **PixelArt** | **LCM**    |
> |-------------------|-----------|-----------|---------------|----------------|--------------|------------|
> | DIRE-ImageNet     | 63.32     | 70.5      | 70.74         | 74.43          | 69.76        | 81.65      |
> | DIRE-SD           | 75.22     | 80.06     | 80.77         | 73.29          | 64.24        | 88.07      |
> | NPR-AIGC          | 76.1      | 71.6      | 73.5          | 73.3           | 87.1         | 76.6       |
> | NPR-GenImage      | 70.9      | 77.2      | 77.9          | 68.9           | 65.1         | 53.7       |
> | AEROBLADE         | 90.81     | 96.48     | 94.03         | 71.53          | 87.84        | 89.99      |
> | Corvi             | 97.87     | 94.81     | 95.32         | 90.93          | 94.16        | 96.15      |
> | Ours          | 99.32     | **98.37** | 97.97         | 98.17          | 98.38        | **99.84**  |
> | Ours-Sync     | **99.4**  | 98.3      | **98.12**     | **98.54**      | **98.58**    | 99.79      |
>
>
> Based on the results, we would like to make two major observations,\
> **a.** Our method outperforms the newly added baselines comprehensively.\
> **b.** The AEROBLADE baseline that we compare against vastly outperforms the DIRE baseline recommended by the reviewer. Given that both of these baselines belong to the paradigm of reconstruction-based detection, we hope this proves that the detectors we compare against are not simple detection techniques, contrary to the reviewer’s suggestion.\
> **c.** Furthermore, we would like to point out that the NPR baseline is also a fine-tuned ResNet-50 based detector. The baseline we select in that paradigm (Corvi), comprehensively outperforms the NPR baseline which once again highlights the fact that the baselines we selected are not simple techniques and in-fact outperform the baselines suggested by the reviewer.
>
> **6. Consider adding ViT based methods**\
> We again would like to highlight that the request to see ViT based models was not present in the original review. We are running some experiments using the ViT backbone. However, since this request by the reviewer was given just 4 days before the deadline, we are not sure if the model will finish training before the discussion period ends. In the case we do get the results, we will try to add them here in the comments section.

---

> > ### Author Response · Authors · 2024-11-24
> > **Authors Response (References)**
> >
> > **References**\
> > [1]Tan, Chuangchuang, et al. "Rethinking the up-sampling operations in cnn-based generative network for generalizable deepfake detection." Proceedings of the IEEE/CVF Conference on Computer Vision and Pattern Recognition. 2024.\
> > [2]Wang, Zhendong, et al. "Dire for diffusion-generated image detection." Proceedings of the IEEE/CVF International Conference on Computer Vision. 2023.\
> > [3]Zhong, Nan, et al. "Patchcraft: Exploring texture patch for efficient ai-generated image detection." arXiv preprint arXiv:2311.12397 (2024): 1-18.\
> > [4]Zhu, Mingjian, et al. "Genimage: A million-scale benchmark for detecting ai-generated image." Advances in Neural Information Processing Systems 36 (2024).\
> > [5]Dhariwal, Prafulla, and Alexander Nichol. "Diffusion models beat gans on image synthesis." Advances in neural information processing systems 34 (2021): 8780-8794.

---

### Official Review · Reviewer_hbmp · 2024-11-03

**Soundness:** 3
**Presentation:** 4
**Contribution:** 3
**Rating:** 6
**Confidence:** 5

**Summary:**

This paper is rooted in a key observation that training a naive binary classifier to detect real/fake might learn the spurious correlations, and thus cannot achieve a more robust detection result. To achieve the alignment of the training dataset, they propose a new dataset by using LDM’s autoencoder for reconstruction purposes. The authors also provide clear evidence and experiments to verify their observation.

**Strengths:**

1. The paper is well-written and has a very clear research motivation.

2. The dataset alignment is a critical problem that might cause the model to learn spurious correlation in the fake image detection field.

3. The authors provide reasonable evidence to validate their findings and observations. Especially the effectiveness of downsampled before and after resizing (Table 5), which is insightful to me.

**Weaknesses:**

- A major limitation of this work is the generality of images generated by Latent Diffusion Models (LDM). Does the detector primarily learn LDM-specific fake patterns, rather than a broader range of patterns such as those generated by GANs? Or does it learn only method-specific fake patterns unique to a particular LDM instance, failing to generalize across the LDM family? A more robust evaluation of this generality would be beneficial.

- This paper defines "fake image detection" as "entire image synthesis." However, this scope does not cover other types of fake images, such as deepfakes (e.g., face-swapping) or talking-head videos. A further discussion of the results of these types of fake images would enhance the paper.

- There is limited analysis of pre-trained models in this work. For instance, CLIP and ResNet-50 (trained on ImageNet) may detect fake images differently. Given CLIP’s rich knowledge, it may be less likely to focus on irrelevant patterns. A deeper analysis and discussion of these differences would strengthen the paper.

- What are the advantages of conducting online reconstruction using LDM (as in previous works, such as ref[1]) compared to offline reconstruction (as used in this work)?

- The alignment issue highlighted here is not entirely new; several prior studies, especially in deepfake detection, have addressed this challenge. For example, refs [2] and [3] discuss the importance of alignment for robust detector performance. Ref [2] shows that a resolution gap between real and fake images can bias models toward interpreting higher resolution as real and lower as fake, while ref [3] demonstrates that aligning real and corresponding fake samples in training improves detector robustness. Further discussion of these relevant works would help contextualize this study’s findings on alignment.

ref[1]: DIRE for Diffusion-Generated Image Detection, ICCV 2023.

ref[2]: DF40: Toward Next-Generation Deepfake Detection, NeurIPS 2024.

ref[3]: Transcending Forgery Specificity with Latent Space Augmentation for Generalizable Deepfake Detection, CVPR 2024.

**Questions:**

1. Does the detector primarily learn LDM-specific fake patterns, rather than a broader range of patterns such as those generated by GANs? Or does it learn only method-specific fake patterns unique to a particular LDM instance, failing to generalize across the LDM family?

2. What is the clear and accurate definition of "fake image" in this work?

3. What are the differences between CLIP and Res50 in the shortcut manner?

4. Offline reconstruction and online reconstruction.

---

> ### Author Response · Authors · 2024-11-19
> **Authors' Response**
>
> 1. **Does the model learn patterns specific to a single LDM? Does the model learn Universal patterns that can generalize to GANs?**\
> We demonstrate that the trained detector learns patterns that generalize to a variety of latent diffusion models. This includes models with changes in VAE architecture such as Kandinsky (line 458-459) and models with changes in UNets (lines 423-426) such as Playground, PixelArt as well as commercial models such as Midjourney.
> \
> \
> Furthermore, we would like to clarify that we are not trying to solve the problem of universal fake image detection in this work. While the community emphasizes on solving the problem of universal fake image detection, in this work, we highlight that reliable detection of fake images from a single family of models is not by any means a trivial task and highlight it by showing several failure modes for a lot of existing models such as spurious correlations (**Sec 5.2, Appendix A.2.2**) and limited performance (Tab 1, 2 and Discussion section). We believe that before tackling the problem of Universal Detection, it is important to build reliable detectors for a single class of models. Please refer to our general comment addressed to all reviewers which elaborates on this point.
> 2. **Discussion of deep fakes, talking face synthesis etc.**\
> Just as we mentioned in our response to the previous question, our goal is not to build a general purpose fake image detector that can detect all kinds of fake images. It is to highlight that even in the case of "entire image synthesis", there are many cautions that one should take while training a detector, and we propose a method for that task. We do agree that detecting deep fakes is also important, maybe societally even more important, but for now, that is outside the scope of our work. We will discuss this clearly in the final version of our paper.
> 3. **Limited Analysis of pre-trained models**
> We thank the reviewer for raising this point regarding the extent of our analysis. In order to verify the veracity of our findings across a variety of pre-training techniques, we include 3 more initializations, a ResNet-50 with the DINO initialization, a modified-ResNet with CLIP initialization and a modified FCN [1] trained on MSCOCO for the task of semantic segmentation. The analysis can be found in **Appendix A.2.4** of the updated submission. We observe that, regardless of the architecture or initialization, our method of using a well-aligned dataset consistently yields better performance than using the dataset provided by the Corvi baseline. For instance, when using the CLIP initialization, our detector achieves an AP of 96.86%, compared to the 88.09% AP achieved by the baseline detector when detecting images generated by Playground. Furthermore, we also observe that the performance across different initializations remains roughly the same.
> 4. **Alignment issues are not entirely novel**\
> We agree that alignment is not an entirely new issue, in fact we ourselves have highlighted prior efforts to ensure alignment in lines 212-216. We thank the reviewer for sharing these articles with us. Article [2] highlights a very similar issue. We will include a discussion about it in our revision. However, it is important to note that [2] talks about how the high-resolution real images possess more high-frequency artifacts which are picked up by the detector. Our work on the other hand emphasizes on how the data augmentations used could behave differently on either class due to a lack of alignment, which in turn can induce spurious patterns that are picked up by the detector.
> \
> \
> Article [3] shared by the reviewer demonstrates the benefit of latent space augmentation in training deep fake detectors. The method involves learning feature extractors specific to each deep fake technique and distilling the learned knowledge into a single student model which serves as a deep fake detector. In order to extend it to detect unseen deep fakes, the authors introduce augmentations in the latent space (such as interpolations) and distill this knowledge to the student as well. While they train on real images and their “deep fake” counterparts, the work does not analyze the effectiveness of data set alignment to the best of our knowledge; we thereby request the reviewer to provide further clarity regarding the same.
> 5. **What is the definition of fake image as per this work?**\
> In this work, we define a fake image as whole-image synthesis by a neural network. We treat any image that was produced by a neural network and then post-processed as fake images as well. Any images that were not produced by neural networks (photographs taken with a camera, graphics programs) as real images.
>
> **References**\
> [1] Long, Jonathan, Evan Shelhamer, and Trevor Darrell. "Fully convolutional networks for semantic segmentation." Proceedings of the IEEE conference on computer vision and pattern recognition. 2015.

---

> > ### Author Response · Authors · 2024-11-19
> > **Authors' Response (continued)**
> >
> > 6. **Advantages of online reconstruction vs offline reconstruction**\
> > We believe that there are no advantages of using online reconstructions in-comparison with offline reconstructions. The predominant factor that motivates online detection is the training data efficiency, however we demonstrate that methods performing online reconstruction such as AEROBLADE (which is similar to the paper referenced by the reviewer), are not robust to post-processing operations as well as subtle changes in the weights of the VAE. We demonstrate this by experimentation in section 5.4 (lines 430-431).
> > 7. **What are the differences between CLIP and ResNet-50 in the shortcut manner?**\
> > We are not entirely clear what the reviewer means by “shortcut manner for CLIP and ResNet-50”, we have not used a CLIP backbone in the initial submission. Therefore, it would be helpful if the reviewer would be more clear regarding the same. However, we have observed that a detector based on CLIP linear probing (Ojha-LDM), learns a very similar spurious correlation to the Corvi method. A subtle difference comes from the fact that in the case of the spurious correlation learned by Ojha-LDM, it predominantly arises from one class being saved in a particular manner (real class has been downsized), therefore the model tends to associate any form of downsizing with real images, we analyze this in detail in **Appendix A.2.2**. In the case of Corvi (ResNet-50) the spurious pattern is induced by how both the classes are augmented in very different ways (due to the misalignment in data). The precise details are present in Section 5.2.

---

> > > ### Author Response · Authors · 2024-11-23
> > > **Authors' Response**
> > >
> > > Please let us know if our rebuttal addresses your concerns, and if we could provide further clarifications that would strengthen your approval of the paper.

---

### Official Review · Reviewer_uw3N · 2024-11-04

**Soundness:** 3
**Presentation:** 3
**Contribution:** 3
**Rating:** 6
**Confidence:** 4

**Summary:**

This paper aims to detect fake images by improving the aspect of data alignment. The paper reconstructs all real images using the autoencoder of a latent diffusion model and then trains fake image detectors to distinguish between real and reconstructed images. The paper claims that this approach prevents the fake image detector from focusing on unwanted artifacts such as semantic content, resolution, or file format. Additionally, they argue that this method of data collection is more efficient than prior works which pair real images with generated images, as it avoids the expensive denoising process associated with generating fake images. The paper shows stronger results than prior works in detecting latent diffusion models and provides an analysis of potential pitfalls in prior works, such as detection bias stemming from resolution differences or compression artifacts between real and fake data.

**Strengths:**

1. The paper proposes a simple and efficient way to collect data for fake image detection.
2. The paper provides an analysis of common pitfalls of prior works, such as image resolution or compression artifacts.

**Weaknesses:**

1. It is unclear whether or not the proposed method would generalize well to architectures other than latent diffusion models. For example, would this generalize well to models such as VQ-VAE[1], VQ-GAN[2], or more recent models with very different autoencoder architectures? The paper mentions that it does not work well on models with vastly different architectures (e.g., FLUX.1-dev), but it is not clear to what extent of architectural changes the proposed method is robust to.
2. The proposed method, by design, disregards semantic content. While the paper claims that this is a desired property, in some contexts, the semantic content may matter. With proper techniques, the autoencoder artifacts may be decimated [3]. In such cases, the proposed method may fail to properly distinguish fake and real. However, prior data-driven approaches may be able to pick up on semantic cues such as over-saturation or contrast, malformed objects, or unrealistic sceneries to detect generated images.
The performance differences after post-processing shown in Table 3, between detectors trained on natural images and shader images, may hint at the importance of semantics.
3. The paper lacks a detailed analysis of sensitivity to various transformations. The paper claims that their method of training detectors is more robust as it forces the detectors to focus more on autoencoder artifacts. However, the experiments do not adequately demonstrate this property, as the paper only compares the performance of a single randomized post-processing technique in Tables 2 and 3. Including a detailed, separate analysis of the robustness to various transformations, similar to those shown in Figure 2 and the figure on page 10 would better illustrate this characteristic and provide better insights for future research.
   - The figure on page 10 should be properly numbered and referenced in the main text.
4. Accuracy is not the best metric for showing the performance of the detectors, as it is sensitive to thresholds. I recommend the authors to consider threshold-less metrics such as average precision (AP). For example, in Tables 1 and 2, applying post-processing actually improves the accuracy of certain detectors (e.g., Cozzolino-LDM). Using threshold-less metrics such as AP may provide a more accurate picture of performance degradation caused by post-processing.

I will give borderline/weak accept for this round and update the rating after the discussion.

&nbsp;

***Minor comments:***
- The figure on page 10 lacks proper numbering.
- While the writing is clear, some of the writing seems less professional. Some examples:
    - Lines 026-027: "Just how effective ... can be"
    - Lines 121-122: "report limitations that even our method cannot circumvent"
    - Lines 274-275: "See how much more efficient our method ... is"
    - Lines 337-338: "Synthesizing a lot of images ..."
    - Lines 517-518: "This is where the promise of ... come"
    - I believe the writing quality could be improved if some of these writings were delivered in a more professional tone (e.g., "a large number of images" instead of "a lot of images").

&nbsp;

[1] Van Den Oord, Aaron, and Oriol Vinyals. "Neural discrete representation learning." Advances in neural information processing systems 30 (2017).

[2] Esser, Patrick, Robin Rombach, and Bjorn Ommer. "Taming transformers for high-resolution image synthesis." In Proceedings of the IEEE/CVF conference on computer vision and pattern recognition, pp. 12873-12883. 2021.

[3] Davide Cozzolino, Giovanni Poggi, Riccardo Corvi, Matthias Nießner, Luisa Verdoliva. "Raising the Bar of AI-generated Image Detection with CLIP". Proceedings of the IEEE/CVF Conference on Computer Vision and Pattern Recognition (CVPR) Workshops, 2024, pp. 4356-4366

**Questions:**

1. What is the rationale behind calibrating the threshold in Table 3? Why is this different than prior experiments (Tables 1 and 2), and what happens if you use the default threshold (0.5) similar to Tables 1 and 2?
2. I'm curious what the authors' intuition behind why the models trained on shaders dataset show more sensitivity to post-processing than the ones trained on natural images.
3. The Appendix A.2.1. shows that the proposed method can train the detector in a more data-efficient manner. I'm curious why this is the case. Is it because it only has to focus on single artifacts?

---

> ### Author Response · Authors · 2024-11-19
> **Authors' Response**
>
> 1. **Generalization to different architectures**\
> In our paper, we show that a detector trained on LDM VQVAE reconstructions can generalize to Stable Diffusion models that use different VAE’s such as Playground, PixelArt etc (Sec 5.4). Our detector can generalize to a completely different MoVQGAN [1] used by Kandinsky (line 458-459). We also show that our detector is more robust to vast changes in the UNet of the diffusion model (lines 423-426). If, however, the main concern of the reviewer was whether our model can work on images generated by the original VQGAN etc (without them being used with diffusion models), then please refer to the general comment about how our goal is different from that of building a universal fake image detector.
> 2. **Proposed method disregards semantic content as a result could be vulnerable to artifact decimation**\
> In the article referenced by the reviewer, the authors conduct an experiment where they down-sample the fake images with a 4x factor. Such an operation could remove some artifacts while approximately preserving the original image. We agree that downscaling an image too much might reduce the artifacts introduced by the autoencoder. In fact, in Figure 2 of the main paper, we do see that the probability of an image being fake drops when the downscaling factor reaches x4 (at 0.25). However, we believe that this is not because there are no artifacts to be found, but rather because that downscaling of x4 was never seen during training. The maximum resolution of our training images are 640 x 640, which can, in very small fraction of cases, can get resized to 256 x 256, i.e., a downsizing factor of x2.5. And since CNNs are not scale invariant, the learnt filters won't behave the same for smaller images.
> Moreover, we also believe that almost every method is going to struggle when images are downscaled by that amount. In fact, going back to Figure 2, we see that the Corvi baseline, which was in fact trained in the standard manner (using fake images with semantic anomalies), struggles even more when fake images are downscaled by x4. In summary, we agree with the reviewer that huge amounts of up/downsizing is a challenge for fake image detection; but we believe it has more to do with a CNN's implicit biases/limitations, and less to do with our specific method. Finally, we would like to clarify that our detector learns patterns beyond low-level traces which the article referenced by the reviewer mentions. This can be inferred based on our model's robustness to various post-processing operations (**Appendix A.2.5**).
>
> 3. **Sensitivity to various transformations**\
> While it is true that we wanted to investigate the resizing issue more deeply because there is a principled reason why Corvi struggles with it, in our evaluations, we have shown robustness to many other randomized post-processing artifacts. Table 2 is for that purpose, where we show the sensitivity to blur, JPEG compression and color jitter operations (lines 412-413). However, we also agree with the reviewer that a more comprehensive study of our method's sensitivity to isolated post processing effects will be helpful. For that purpose, we have conducted additional tests separately analyzing the model's robustness to blur, Gaussian noise, and JPEG compression. We refer the reviewer to **Appendix A.2.5** for the analysis. Particularly in the case of blur, we notice that blurred fake images are more confidently classified as fake using our method compared to Corvi.
>
> 4. **The figure in page 10 should be properly numbered and referenced**\
> We have numbered the figure properly (currently Fig 5) and referenced it in the updated version of the paper.
>
> 5. **Using thresholdless metrics**\
> We agree that using other threshold-less metrics can be helpful in addition to accuracy. As a result, we have evaluated all the approaches by using Average Precision (AP) and true positive rate at 5 false positive rate (tpr5fpr). The results can be found in **Appendix A.2.3** of the updated version. We achieve a very high AP and TPR@5FPR. For instance, when detecting Playground-generated images, we achieve an AP of 98.54% and a tpr@5fpr of 92.88%, compared to the next best method (Corvi), which achieves an AP of 90.93% and a tpr@5fpr of 65.75% on the same task.
>  The reason we used accuracy as the metric was because we wanted to simulate a real-life testing environment where calibrating the detector on the test dataset will not be possible. This implies that having a good test AP is not always useful in practice.
>
> 6. **Minor Comments/Suggestions**\
> We thank the reviewer for providing us with feedback on the writing. We have made the appropriate changes and have highlighted it in red in the updated version of the paper.
>
> **References**\
> [1] Zheng, Chuanxia, et al. "Movq: Modulating quantized vectors for high-fidelity image generation." Advances in Neural Information Processing Systems 35 (2022): 23412-23425.

---

> ### Author Response · Authors · 2024-11-19
> **Authors' Response (continued)**
>
> 7. **Why calibrate the threshold only for the shaders experiment?**\
> We conduct the shaders experiment to show the following-by using a well-aligned dataset, we can detect fake images coming from completely unrelated domains (which is very relevant given the internet scale datasets that are used to train models). Since, the model does not observe any natural images during training, it is not likely for those images to conform to a 0.5 threshold, therefore, we calibrate the threshold. Note that, we calibrate the threshold based on a validation set, and we calibrate it for the Corvi/Ours-Sync methods in Table 3 in order for fair comparison. The details of the experiment are given in **Appendix A.1.2**. If we do not calibrate the threshold, the accuracies of the models drop a bit. Please note that the calibrated thresholds of our models are around 0.44 for the Ours version and 0.6 for the Ours-Sync version trained on shaders.
> 8. **Why do shaders struggle with post-processing?**\
> We have two hypotheses for this. First, the images from Shaders dataset have a resolution of 384x384, whereas those in the natural case (LSUN + MS COCO) have images ranging from 256x256 to 640x640 (line 718-723). Due to the wide range of resolutions present, we think that the detector trained on natural images encounters a wider range of resizing operations during training which results in increased robustness. In Table 2, resizing is one of those post-processing operations that is randomly applied to test images (to simulate real world processes), to which the detectors trained on natural images could be more robust towards.
> Our second hypothesis is that both of our detectors, one trained on natural and the other on Shaders, won't be perfect in their reconstructions. For example, [2] showed that 4-channel LDM autoencoders struggle with reconstructing text. And hence, the resulting detector might learn to associate 'good-looking text' to real and 'gibberish' to fake. Similarly, the same autoencoder will struggle in reconstructing Shaders images as well, and the detector might consequently learn to associate those features to real/fake images. However, in the first case, the features learnt (good text -> real) can actually be useful at test time to detect fake images as fake (e.g., when a test image as gibberish text), whereas those learnt by Shaders inconsistencies will likely not transfer to the natural image test data.
>
> 9. **Why is the detector trained on our approach more data-efficient?**\
> We believe that this is a direct effect of dataset alignment. In the case of Corvi, with a small dataset which is not well-aligned, there could exist multiple subtle spurious patterns (e.g. a real black cat vs a fake white cat). The model can overcome such patterns with more data. Our detector on the other hand will not capture such shallow, spurious patterns since the dataset is properly aligned. This increases the likelihood of our model to capture deeper, more common patterns that make an image fake.
>
> **References**\
> [2] Dai, Xiaoliang, et al. "Emu: Enhancing image generation models using photogenic needles in a haystack." arXiv preprint arXiv:2309.15807 (2023).

---

> > ### Author Response · Authors · 2024-11-23
> > **Authors' Response**
> >
> > Please let us know if our rebuttal addresses your concerns, and if we could provide further clarifications that would strengthen your approval of the paper.

---

> > > ### Comment · Reviewer_uw3N · 2024-11-25
> > >
> > > I believe the authors addressed most of my concerns. I have a few more comments:
> > >
> > > 1. (Q1) My intuition is that since the paper's approach is to detect autoencoder artifacts, I suspect it would perform well on VQ-VAEs or VQ-GANs, as the distance from LDMs to those methods is much closer than those of FLUX. It would be interesting to see such *spectrum* of architectures that this method can detect, with x-axis possibly being the year they are released and y-axis being the performance. This will provide a much more interesting picture of how generalizable this approach is outside of LDM-like architectures. While this analysis may not be possible to be conducted within the rebuttal timeframe, I suggest the authors to consider this in future versions.
> > > 2. (Q7) The reason for threshold calibration doesn't seem to be sufficiently detailed in the paper. I suggest the authors to discuss this reasoning in the paper to further improve the clarity, especially since it is different from earlier experiments.
> > > 3. I resonate with Reviewer **caUt** that Sec. 5.3 (Computational Efficiency) seems more fitting for the appendix. While interesting, this analysis is distant from all other experiments conducted throughout the paper. Rather, I think Appendix A.2.1 (Effect of Dataset Size) is more suitable to be moved to the main text as it illustrates another benefit of avoiding spurious correlations.
> > > 4. I also agree with Reviewer **ecmv** that the paper's title should clearly show the scope of this paper (detecting LDM-like architectures). It is easy to confuse from the title that the paper is targeting a more general fake image detector, rather than only LDM.
> > >
> > > I believe the paper is worthy of publication as it provides interesting analyses of the importance of data in detecting fake images. The research community could benefit from the insights provided by this paper. I recommend the authors to further improve the clarity of the paper, especially in experiments and the scope of the paper.

---

> > > > ### Author Response · Authors · 2024-11-25
> > > > **Authors' Response**
> > > >
> > > > We thank the reviewer for taking the time to provide constructive feedback, and for stating that our paper is worthy of being published.
> > > >
> > > > As mentioned in our response to reviewer ecmv, we will add qualifiers to the title to ensure it appropriately represents the scope of our work in the final version of the paper. Additionally, we will elaborate on our motivations for calibrating the threshold in the shaders experiment and move the computational efficiency section to the Appendix, allowing us to include more critical aspects of our analysis in the main body of the paper.

---

### Author Response · Authors · 2024-11-19
**Authors' Response**

We thank the reviewers for their time in giving us constructive feedback. It is encouraging that the reviewers found our analysis of the effects of scaling/compression operations in the context of fake image detection insightful (hbmp, caUt) and believe it will be valuable (uw3N) to the community. The reviewers also found the proposed solution to be interesting (caUt), simple (uw3N, ecmv) and efficient (uw3N, ecmv). Overall, the reviewers found the work to be well-written (hbmp, ecmv) with sufficient experiments to support the claims (hbmp, ecmv).

Many reviewers raised questions about the effectiveness of our approach in detecting images from other kinds of generative models (like GANs). We want to provide a common clarification to that question: our goal is not to build a universal fake image detector that can detect a fake image from all possible breeds of generative models. Rather, our goal is to first highlight that even the more simple task - robustly detecting fake images within the same breed - is more difficult than previously thought, due to the presence of spurious features in the dataset, such as resizing artifacts (lines 290–302), which can be used to separate real and fake images and we propose a method to address this issue. Within that breed (LDM), there are many variants of generative models out there (e.g., changes in the UNet, VAE architecture). And we show that our method is indeed effective in detecting images from all of them; see Tables 1, 2. In particular, we show improvements in accuracy on detecting fake images over the state-of-the-art method of Corvi on recent latent diffusion models such as Playground, PixelArt, and Midjourney by +52.09%, +22.59%, and +21.91%, respectively. We will further clarify this distinctive goal in the final version of our paper.

Furthermore, we have addressed several important concerns raised by the reviewers in the Appendix of the revision,
1. **Threshold-agnostic metrics**: We report the Average Precision (AP) and True Positive Rate at 5% False Positive Rate (tpr@5fpr) in Appendix A.2.3. We achieve a very high AP and TPR@5FPR. For instance, when detecting Playground-generated images, we achieve an AP of 98.54% and a tpr@5fpr of 92.88%, compared to the next best method (Corvi), which achieves an AP of 90.93% and a tpr@5fpr of 65.75% on the same task.
2. **Impact of Architecture/pretraining**: We show that our approach has benefits in a manner which is agnostic to the network architecture/initialization. The results are present in Appendix A.2.4. The experiments demonstrate that the advantages of our approach persist across various architectures and initializations. For instance, when training a CLIP-initialized Modified ResNet backbone, we observe that training on our well-aligned dataset achieves an AP of 99.36% vs. 91.02% achieved using the dataset provided by Corvi, on images generated by the Latent Consistency Model.
3. **Robustness to additional post-processing operations**: We study the sensitivity of our detector to blurring, additive noise and JPEG compression in Appendix A.2.5. Experiments show that training on our well-aligned dataset demonstrates increased robustness to blurring compared to training on the dataset provided by Corvi.
4. We have also properly labeled the Figure in page 10, and referenced it in the main paper.

For convenience, we have used red, to highlight the text which has been changed/added to the original version in our new version.

Additionally, we address several of the reviewer specific concerns below.

---

### Public Comment · ~Xiaoxuan_Yao1 · 2024-11-22
**Questions About the Motivation and Innovation of the Proposed Method**

This paper focuses on the effectiveness of dataset alignment in fake image detection. Motivated by controlling the differences of the real and fake images (e.g., resolution, semantic content and color tone) can help the detector pay more attention to subtle artifacts introduced by LDM, the authors propose a method that reconstructs the real images using LDM’s autoencoder. Reconstructed images will be treated as fake images. Then the authors train a detector using the same training recipe as proposed in [1]. However, I have three questions about the motivation and innovation of the proposed method.

1. The idea of reconstructing the real images to preserve the image visual content while leaving the artifacts of the generative model and treating the reconstructions as new fake samples have been proposed in [3][4]. Compared with them, is there any evidence to suggest that reconstruction using LDM’s autoencoder preserves the visual content better?

2. Reconstructing the images using LDM’s autoencoder has been proposed in [2], which is a baseline included in the experiments. Therefore, it is difficult to attribute the performance improvement to dataset alignment since [2] utilizes the same data processing techniques. This improvement is very likely due to the use of the classifier proposed in [1] and the idea mentioned in Q1, considering that [2] just employs a simple threshold-based detection and treats the reconstructions as features of training samples rather than new fake samples.

3. The idea of ensuring that the resolutions of the reconstructions are identical to that of their real counterparts is similar to avoiding the bias introduced by the different resolution of real images and fake images, which has been proposed in [5]. The detrimental impact of resizing on fake image detection has also been mentioned in [1]. However, I highly appreciate the authors’ novel discovery that aligning the resolution of real images and fake images improves detector's robustness to resized images.

Would it be possible for the authors to provide additional clarifications about the motivations and strengths of the proposed methods compared to previous works?

**References**

[1] Corvi, Riccardo, et al. "On the detection of synthetic images generated by diffusion models." ICASSP 2023-2023 IEEE International Conference on Acoustics, Speech and Signal Processing (ICASSP). IEEE, 2023.

[2] Ricker, Jonas, Denis Lukovnikov, and Asja Fischer. "AEROBLADE: Training-Free Detection of Latent Diffusion Images Using Autoencoder Reconstruction Error." Proceedings of the IEEE/CVF Conference on Computer Vision and Pattern Recognition. 2024.

[3] Chen, Baoying, et al. "DRCT: Diffusion Reconstruction Contrastive Training towards Universal Detection of Diffusion Generated Images." Forty-first International Conference on Machine Learning.

[4] Yu, Xiao, et al. "SemGIR: Semantic-Guided Image Regeneration Based Method for AI-generated Image Detection and Attribution." Proceedings of the 32nd ACM International Conference on Multimedia. 2024.

[5] Grommelt, Patrick, et al. "Fake or JPEG? Revealing Common Biases in Generated Image Detection Datasets." arXiv preprint arXiv:2403.17608 (2024).

---

> ### Author Response · Authors · 2024-11-22
> **Authors' Response**
>
> Thank you for your questions. We address the concerns raised below,
>
> **The idea of training on reconstructions has been proposed before in some works, but the motivations are not too similar**\
> 1. **[3] Uses Reconstructions to train a detector, but motivations are different**\
> The goal of authors in [3] was to improve existing fake image detector's generalization ability with the aid of reconstructed real images. However, their motivations for doing so are different from ours. Their goal is to simply make the task of the detector more difficult, which they do by introducing real image reconstructions as an additional type of "harder-to-classify" fake images. But the standard fake images, generated using text by the iterative denoising process, are still part of the overall fake category in the training data. Please refer to Fig. 2 of their paper which shows that the overall fake category is composed of three types of fake images, including the standard fakes and their reconstructions. This implies that the fundamental problem that we have discussed in our work, i.e., any kind of misalignment, e.g., resizing artifacts due to differing original resolutions, can still be a problem for their approach. In contrast, in our work, we wanted to completely align the real and fake image distribution so that, as much as possible, there is very little scope for learning some other spurious features. In fact, if we had to use the method proposed by authors in [3] for the data used by Corvi et al. which uses generated images at 256 x 256 resolution (with real images at a different resolution), then among the three types of fake images (Fig. 2 in [3]), two of them will still have the resolutions of 256 x 256, and will have the same problems as we discuss in Sec. 5.2.\
> \
> Next, the way we do reconstructions is different and much simpler. For the LDM based models (e.g., SDv1.4, which is used as the training data in [3]), we do not need to do any multi-step DDIM inversion, but simply pass the real image through the autoencoder to get its reconstructions. This is much more computationally efficient (Fig. 3; Sec. 5.3). Furthermore, training a detector on DDIM reconstructed images, will be sensitive to the change in the UNet (while preserving the VAE). Ours on the other hand will not be sensitive to changes in the UNet. We have empirically observed this in our work in Lines 423-426. \
> \
> Finally, because we have had such a strong emphasis on making the real and fake dataset well-aligned, we show that the principle can be used to build effective detectors even when we only have access to algorithmically generated images (which do not capture the semantics of natural images), and do not have access to natural real images (Sec. 5.5).
>
> 2. **[4] does not use reconstructions, furthermore we have compared against a similar paradigm in our paper**\
> We would also like to point out that [4] is not very similar to our approach. They use an image-to-text model in order to get a prompt for each real image which can be used to generate the corresponding fake image. In fact, this is very similar to the existing paradigm that we compare against and also acknowledge in our work (refer Fig 1 and line 212-216). The existing paradigm relies on the captions associated with the image in order to generate similar fake images. For the class of latent diffusion models, our proposed approach is more efficient (does not require costly captioning or iterative denoising). Furthermore, using the LDM autoencoder leads to better reconstructions, i.e., more aligned fake images.
>
> **Autoencoder reconstructions are generally more accurate than DDIM reconstructions for LDM’s**\
> The autoencoder reconstruction is among the best approximations to the real image that can be made by the latent diffusion model. Intuitively, DDIM inversion tries to approximate this encoded latent, this results in additional approximation error which could affect reconstruction quality. This has been discussed in several prior works such as [6] and [2]. Importantly, [2] has a clear experiment demonstrating the difference in the reconstruction quality. They observe that using DDIM reconstructions increases the distance between the fake images and their reconstruction and show in conclusion that just using only LDM’s autoencoder for reconstructions achieves better reconstructions. [6] uses the autoencoder reconstruction ability in order to measure the upper bound of generation quality (hypothesis being that autoencoder reconstruction is the closest to real images one can get to). Additionally, DDIM inversion involves numerically solving a forward ODE and a reverse ODE, it is therefore twice as expensive as if we were to generate the images from scratch (similar to Corvi), therefore using autoencoder reconstructions is much more efficient in comparison.

---

> > ### Author Response · Authors · 2024-11-22
> > **Authors' Response (continued)**
> >
> > **The motivations of AEROBLADE [2] for using reconstructions is not to improve dataset alignment, therefore its performance should not be used to reason about the effectiveness of dataset alignment**\
> > We think there might be a confusion regarding what authors in AEROBLADE [2] are doing. Yes, they are using LDM autoencoder's reconstructions. But those reconstructions have very little, if anything, to do with real-fake dataset alignment that we explore in our work. Specifically, the authors in [2] use the following principle - all images generated by LDMs come from a particular latent, whereas the real images do not have one (strictly speaking). Hence, if one reconstructs a LDM generated fake image using the same LDM's autoencoder, the reconstruction should be very close to the input LDM fake image (authors use LPIPS to measure feature similarity), whereas for real images, the reconstruction will be not as close. In other words, to detect whether an image is fake, authors in [2] are simply checking how close it is to its reconstructions. They do not need to have their real/fake images aligned. Whereas in our case, we are training a detector on real images and their reconstructions, with the goal being to mitigate the ability of the detector to learn spurious correlations by eliminating them from the dataset. Furthermore, we have isolated the effectiveness of dataset alignment in several of our experiments by comparing with [1] (which has the same algorithm as our approach, but not an aligned dataset). Therefore, we do not agree that the performance of [2] can be used to determine the effectiveness of dataset alignment.
> >
> > **While [1] mentions the need to avoid resizing, it is not for the reasons that we identify in our paper**\
> > We want to clarify that the motivations are completely different. The authors of [1] motivate it from the perspective of losing out decisive, high-frequency information. They mention that loss in high frequency information could wash away the fake image artifacts. This has nothing to do with misalignment between real and fake images that can occur due to different image resolutions; which is an issue that we identify in our work. In fact, further evidence for the same comes from the fact that we highlight and clearly explain a spurious correlation which has been picked up by the detector trained by [1].
> >
> > **[5] report similar resizing based biases, but we propose a different way to handle it**\
> > [5] has a section talking about biases that could come from misalignment in resolution. Their proposal to fix it is by just selecting real/fake images in a given range so that their resolution differences are minimal. While this can be effective, this would mean that we will need to throw out a lot of real images, something that the authors themselves acknowledge in their paper (Sec. 4.2 - "This data selection in total reduced the number of training samples from approximately 320,000 to 75,000"). Using our method, we do not need to throw out any of the real images, but just simply get the reconstructions for all of them which enables us to have fake images of the same resolution in a one-to-one manner. Furthermore, while the observation made by the authors in [5] was indeed helpful, it still needs an effort on our side to study all the ways in which the real and fake images are different, and manually account for them (if we can). In contrast, using the reconstruction technique described in this work, we can be a bit more relaxed since the autoencoder reconstructions will, for the most part, take care of many types of alignment issues (e.g., color, contrast, size, blur). In fact, we show that not only do we get greater robustness to spurious correlations with respect to resizing (Sec 5.2), but also to blurring (Appendix A.2.5). Finally, as mentioned in response to comparison to [3], the alignment done by autoencoder reconstructions can even let one train a detector on non-natural real/fake images, and detect natural looking real/fake images (Sec. 5.5).

---

> ### Author Response · Authors · 2024-11-22
> **Authors' Response (References)**
>
> **References**\
> [1] Corvi, Riccardo, et al. "On the detection of synthetic images generated by diffusion models." ICASSP 2023-2023 IEEE International Conference on Acoustics, Speech and Signal Processing (ICASSP). IEEE, 2023.\
> [2] Ricker, Jonas, Denis Lukovnikov, and Asja Fischer. "AEROBLADE: Training-Free Detection of Latent Diffusion Images Using Autoencoder Reconstruction Error." Proceedings of the IEEE/CVF Conference on Computer Vision and Pattern Recognition. 2024.\
> [3] Chen, Baoying, et al. "DRCT: Diffusion Reconstruction Contrastive Training towards Universal Detection of Diffusion Generated Images." Forty-first International Conference on Machine Learning.\
> [4] Yu, Xiao, et al. "SemGIR: Semantic-Guided Image Regeneration Based Method for AI-generated Image Detection and Attribution." Proceedings of the 32nd ACM International Conference on Multimedia. 2024.\
> [5] Grommelt, Patrick, et al. "Fake or JPEG? Revealing Common Biases in Generated Image Detection Datasets." arXiv preprint arXiv:2403.17608 (2024).\
> [6] Esser, Patrick, et al. "Scaling rectified flow transformers for high-resolution image synthesis." Forty-first International Conference on Machine Learning. 2024.

---

### Public Comment · ~Dan_Oneata2 · 2024-12-03
**Implications and further references**

This paper provides a sensible way of avoiding shortcuts that are inadvertently introduced when creating fakes; for example, silence ([Müller et al., 2021](https://arxiv.org/pdf/2106.12914)), bitrate information ([Borzì et al., 2022](https://openaccess.thecvf.com/content/CVPR2022W/WMF/papers/Borzi_Is_Synthetic_Voice_Detection_Research_Going_Into_the_Right_Direction_CVPRW_2022_paper.pdf)), preprocessing artefacts ([Chai et al., 2020](https://arxiv.org/pdf/2008.10588)).

**Implications.** However, the main implication of the propsed approach is that the detection model will exclusively focus on the fingerprint of the chosen generative model.
This means that the detection method:
- ... cannot localise inpainted regions by LDM-based methods; as discussed in ([Mareen et al., 2024](https://arxiv.org/pdf/2407.11566); [Smeu et al., 2024](https://arxiv.org/pdf/2409.08849)).
- ... will mark benign manipulations (such as superresolution or image enhancement) as fakes.
- ... will be susceptible to laundering ([Mandelli et al., 2024](https://arxiv.org/pdf/2407.10736)): take a compromising image, pass it through the LDM encoder-decoder and claim that it was fake.

Maybe a discussion on the implications of focusing on the fingerprint (and disregarding the semantics) is useful?

**Further references.** Some other papers that are related and may be worth citing:

- [Chen et al. (2024)](https://openreview.net/pdf?id=oRLwyayrh1) also use reconstruction through diffusion models, although they reproject through the entire model (not only the VAE) and they also reproject the fake samples. (Edit: I now see that this paper was also mentioned in the comment of [Xiaoxuan Yao](https://openreview.net/forum?id=doBkiqESYq&noteId=XsrslAurGB), so sorry for the duplicate).
- In the audio domain, passing the real audio files through the vocoders is a very similar approach and fairly common ([Wang and Yamagishi, 2023](https://arxiv.org/pdf/2210.10570); [Sun et al., 2023](https://openaccess.thecvf.com/content/CVPR2023W/WMF/papers/Sun_AI-Synthesized_Voice_Detection_Using_Neural_Vocoder_Artifacts_CVPRW_2023_paper.pdf); [Wang and Yamagishi, 2024](https://ieeexplore.ieee.org/stamp/stamp.jsp?arnumber=10446331)).

---

References:

- Borzì, S., Giudice, O., Stanco, F., & Allegra, D. (2022). Is synthetic voice detection research going into the right direction?. CVPRW.
- Chai, L., Bau, D., Lim, S. N., & Isola, P. (2020). What makes fake images detectable? understanding properties that generalize. ECCV.
- Chen, B., Zeng, J., Yang, J., & Yang, R. DRCT: Diffusion Reconstruction Contrastive Training towards Universal Detection of Diffusion Generated Images. ICML.
- Mandelli, S., Bestagini, P., & Tubaro, S. (2024). When Synthetic Traces Hide Real Content: Analysis of Stable Diffusion Image Laundering. WIFS.
- Mareen, H., Karageorgiou, D., Van Wallendael, G., Lambert, P., & Papadopoulos, S. (2024). TGIF: Text-Guided Inpainting Forgery Dataset. WIFS.
- Müller, N. M., Dieckmann, F., Czempin, P., Canals, R., Böttinger, K., & Williams, J. (2021). Speech is silver, silence is golden: What do ASVspoof-trained models really learn?. arXiv preprint arXiv:2106.12914.
- Smeu, S., Oneata, E., & Oneata, D. (2024). DeCLIP: Decoding CLIP representations for deepfake localization. WACV.
- Sun, C., Jia, S., Hou, S., & Lyu, S. (2023). Ai-synthesized voice detection using neural vocoder artifacts. CVPRW.
- Wang, X., & Yamagishi, J. (2023). Spoofed training data for speech spoofing countermeasure can be efficiently created using neural vocoders. ICASSP.
- Wang, X., & Yamagishi, J. (2024). Can large-scale vocoded spoofed data improve speech spoofing countermeasure with a self-supervised front end?. ICASSP.

---

> ### Author Response · Authors · 2024-12-04
> **Authors' Response**
>
> Thank you for your questions, we address the concerns below,
>
> **Inpainting**\
> In this work, we restrict our scope to “entire-image synthesis”. As mentioned in our response to reviewer hbmp, we will discuss this clearly in our final version.
>
> **Will mark benign manipulations as fake**\
> Our problem statement focused on detecting any type of fake image generated by a specific family of generative models. Consequently, as long as super-resolution or enhancement is performed by a latent diffusion model (similar to Stable Diffusion), our method will classify it as fake. We emphasize that any image enhanced or restored by a latent diffusion model is, at its core, manipulated by a neural network. Our aim is to detect such images as fake regardless of whether they are malicious or benign.
>
> **Susceptibility to Laundering**\
> If a real image (such as a compromising photo) can be reproduced by the generative model, we believe that the generated image should be detected as fake. Practically, this is driven by the goal of detecting fake images that are visually plausible and closely resemble realistic ones. Regarding the presented scenario, we want to emphasize that as long as the original image (which is not fake) has not been altered, it should be identified as real by a model designed to focus on the generator’s artifacts.
>
> **Citations**\
> We will cite relevant work such as DRCT and explain the differences in our final version.

---

### Meta-Review · Area_Chair_95fP · 2024-12-22

**Metareview:**

Summary.
This paper aims to detect fake images by improving the aspect of data alignment. The paper reconstructs all real images using the autoencoder of a latent diffusion model and then trains fake image detectors to distinguish between real and reconstructed images. The paper claims that this approach prevents the fake image detector from focusing on unwanted artifacts such as semantic content, resolution, or file format.

Strengths.
The paper proposes a simple and efficient way to collect data for fake image detection.
The paper is well-written and has a clear research motivation.

Weaknesses.
Proposed method disregards semantic content as a result could be vulnerable to artifact decimation.
The proposed method uses a detector for a specific decoder (specifically based on latent diffusion models). The generalization of such a detector is unclear.
The proposed method seems to be sensitive to different transformations.
The alignment issue highlighted here is not entirely new.

The following paper also uses reconstructions to train a detector. Authors argued that the motivations are different, but I see a strong similarity. In fact, the Chen et al. paper is more focused and accurately represents the detection of diffusion-generated images, whereas this paper (while restricted to latent diffusion models) presented a similar method without qualifying its limitations.

Chen, Baoying, et al. "DRCT: Diffusion Reconstruction Contrastive Training towards Universal Detection of Diffusion Generated Images." Forty-first International Conference on Machine Learning.


Missing.
It is unclear if the proposed method can detect fake images from any arbitrary generator or is it restricted to one (or a small subset) of generative models that have similar artifacts as the trained decoder.

Reasons for reject.
The idea of dataset alignment is discussed in other papers (e.g., Chen et al. and other papers mentioned by reviewer hbmp). The arguments authors provided on differences are not very convincing.
It is unclear if the proposed method generalizes to arbitrary generators or latent models with autoencoders that substantially differ from the one used in training. Detailed discussion and analysis are warranted to accept the claims in the paper.

Reasons for accept.
All the reviewers are leaning toward accept despite their reservations about the claims and scope of the paper.

**Additional Comments On Reviewer Discussion:**

This paper had a long discussion between authors and reviewers.

Reviewers raised questions about the scope of detection limited to latent diffusion models, generalizability of the method, lack of novelty and technical contributions, limited robustness tests, and clarity of the experiments.

Authors provided detailed response and additional results.

Overall, this paper is a real borderline case. Reviewers are leaning towards accept but they have their reservations about the claims and scope of the paper. I have read all the reviewer comments and public comments; I am concerned about the claims/scope of the paper, novelty of the alignment idea, and generalizability of the proposed method to fake image detection.

---

### Decision · Program_Chairs · 2025-01-22

Accept (Poster)